# Engineering T cells to enhance 3D migration through structurally and mechanically complex tumor microenvironments

Erdem D. Tabdanov [1,2,3,13✉], Nelson J. Rodríguez-Merced [1,2,13], Alexander X. Cartagena-Rivera [4], Vikram V. Puram [1,2,5], Mackenzie K. Callaway [1,2], Ethan A. Ensminger [1,2], Emily J. Pomeroy [6,7,8], Kenta Yamamoto [6,7,8], Walker S. Lahr [6,7], Beau R. Webber [6,7,8,9], Branden S. Moriarity [6,7,8,9], Alexander S. Zhovmer [10,12] & Paolo P. Provenzano [1,2,6,9,11✉]

Defining the principles of T cell migration in structurally and mechanically complex tumor microenvironments is critical to understanding escape from antitumor immunity and optimizing T cell-related therapeutic strategies. Here, we engineered nanotextured elastic platforms to study and enhance T cell migration through complex microenvironments and define how the balance between contractility localization-dependent T cell phenotypes influences migration in response to tumor-mimetic structural and mechanical cues. Using these platforms, we characterize a mechanical optimum for migration that can be perturbed by manipulating an axis between microtubule stability and force generation. In 3D environments and live tumors, we demonstrate that microtubule instability, leading to increased Rho pathway-dependent cortical contractility, promotes migration whereas clinically used microtubule-stabilizing chemotherapies profoundly decrease effective migration. We show that rational manipulation of the microtubule-contractility axis, either pharmacologically or through genome engineering, results in engineered T cells that more effectively move through and interrogate 3D matrix and tumor volumes. Thus, engineering cells to better navigate through 3D microenvironments could be part of an effective strategy to enhance efficacy of immune therapeutics.

[1] Department of Biomedical Engineering, University of Minnesota, Minneapolis, Minnesota, USA. [2] University of Minnesota Physical Sciences in Oncology Center, Minneapolis, Minnesota, USA. [3] Department of Pharmacology, Penn State College of Medicine, Hershey, Pennsylvania, USA. [4] Section on Mechanobiology, National Institute of Biomedical Imaging and Bioengineering, National Institutes of Health, Bethesda, Maryland, USA. [5] University of Minnesota Medical School, Minneapolis, Minnesota, USA. [6] Masonic Cancer Center, University of Minnesota, Minneapolis, Minnesota, USA. [7] Department of Pediatrics, University of Minnesota, Minneapolis, USA. [8] Center for Genome Engineering, University of Minnesota, Minneapolis, MN, USA. [9] Stem Cell Institute, University of Minnesota, Minneapolis, Minnesota, USA. [10] National Heart, Lung, and Blood Institute, National Institutes of Health, Bethesda, Maryland, USA. [11] Institute for Engineering in Medicine, University of Minnesota, Minneapolis, Minnesota, USA. [12] Present address: Center for Biologic Evaluation and Research, US Food and Drug Administration, Silver Spring, Maryland, USA. [13] These authors contributed equally: Erdem D. Tabdanov, Nelson J. Rodríguez-Merced. ✉email: ekt5171@psu.edu; pprovenz@umn.edu

Although chemical signals have an important role in attracting T cells into solid tumors, the physical features of the stroma (e.g., architecture and mechanics) also strongly influence T-cell infiltration and their ability to effectively distribute throughout, and sample, the entire tumor mass. Indeed, the complex stromal reaction in solid tumors can limit access and effective distribution of T cells creating antitumor immunity-free sanctuaries[1–5]. Furthermore, many solid tumors are rich with aligned extracellular matrix (ECM) networks[6–8], which provide contact guidance for carcinoma cells[6,9–11], and can also direct migration of infiltrated T cells in solid tumors[1,2,12]. These issue are particularly strong in pancreatic ductal adenocarcinomas (PDAs) that have a robust fibrotic and immunosuppressive stroma[13–16], and are frequently characterized by very limited and/ or heterogeneous distributions of cytotoxic T cells[4,16–18]. Yet, our understanding of how native and engineered T cells migrate through mechanically complex tumor microenvironments (TMEs) is quite incomplete. Although considerable efforts have been made to better understand the biophysics of the immune synapse[19,20], much less is known about the regulation of T-cell migration in physically complex microenvironments. However, elucidating migration behavior and employing rational engineering design approaches, with genome and cell engineering, to alter native cytotoxic T cells and/or further enhance engineered T cells so that they can most effectively migrate through and sample the entire tumor volume, can enhance therapeutic efficacy against solid tumors and indeed further improve cell-based cancer therapeutics in general.

To further develop design criteria for enhancing T-cell migration in tumors, it is critical to first understand the states that result in more robust T-cell motility. For instance, T-cell three-dimensional (3D) motility in healthy and tumor tissues can be influenced by a dynamically balanced superposition of phenotypes, such as an amoeboid phenotype with low adhesion pseudopodia and a more mesenchymal-like phenotype with adhesive spreading, a phenomenon known as amoeboid-mesenchymal plasticity[21–23], where adaptive phenotypic switching can be advantageous for navigating heterogeneous cell and ECM conditions. As such, defining the key principles of the amoeboid-mesenchymal plasticity balance in T cells during normal tissue scanning can increase our understanding T-cell motility. Thus, platforms that capture key physical features of ECM microenvironments, induce relevant phenotypes, drive phenotypic switching, or allow for co-existent phenotypes are critical. For instance, although fundamental advances in our understanding of T-cell motility have derived from two-dimensional (2D) systems[24] and 3D microchannels[25,26], to date, existing platforms have not captured tumor-relevant ECM architectures, such as aligned fiber networks with variable stiffness[6,8,11]. Indeed, although flat 2D systems have had great utility for studying T-cell motility[27,28], they lack the capacity to facilitate steric cell–microenvironment interactions from textured and aligned ECMs, which can be associated with coexisting amoeboid and mesenchymal phenotypes[11], and are distinct from microchannels where locomotion takes place under uniform spatial confinement[26]. Thus, engineered platforms that incorporate disease-relevant ECM architecture, with variable stiffness, and allow high-throughput quantitative analysis of motility for large numbers of cells are needed to dissect out the fundamental drivers of diverse T-cell migration modes and provide design criteria for immune cell engineering.

Here we hypothesize that by elucidating mechanisms governing T-cell motility behavior in response to defined mechanical and architectural microenvironmental cues, we can further utilize these principles to rationally perturb T cells, to enhance their migration through physically complex TMEs. We use engineered platforms to elucidate fundamental modes of T-cell migration governed by ECM architecture and mechanics, and establish that stiffness-dependent T-cell guidance along nanogrooves that mimic tumor-relevant aligned ECM is greatly enhanced by sterically interactive amoeboid behavior that competes against a coexisting mechanically controlled mesenchymal-like mode, which we define as having lamellipodium-driven flat adhesive spreading. This competition results from the redistribution of forces between high cortical contractility in the amoeboid state and greater cell-substrate interaction forces associated with the mesenchymal-like state. By disrupting microtubule (MT) dynamics, we can drive cells into a more amoeboid phenotype that shows enhanced migration speed and directionality on "2.5D" nanotextured environments, while MT stabilization decreases both the migration speed and directionality. We test our findings in engineered 2D to 3D systems and in live tumor PDAs. Consistent with findings in "2.5D" engineered systems, disruption of MTs enhances migration in 3D and tumor environments, and we demonstrate that this results from increased cell contractility in the form of high cortical contractility associated with amoeboid behavior. This MT–contractility axis can be confirmed and captured by using CRISPR (clustered regularly interspaced short palindromic repeats) technology to genome engineer T cells lacking *GEF-H1* (*ARHGAP2*), a Rho guanine exchange factor that can increase Rho activity following MT destabilization. Knockout (KO) of GEF-H1 in the context of MT instability abrogates the increased migration response, whereas GEF-H1 without MT disruption significantly enhances migration. Thus, T cells can be specifically perturbed or engineered to more effectively navigate through complex TMEs, which could become part of an effective strategy to more efficiently sample tumor volumes, to maximize the impact of T-cell-related immunotherapeutic approaches.

## Results

**Mechanically regulated interactions with cancer-relevant architectures.** Aligned stromal ECM in tumors, which has also been reported to guide tumor-infiltrating T cells[1,2], contains anisotropic nanostructured collagen with spacing and textures that can be effectively mimicked through nanotexturing technology[8,11]. This allows us to impart relevant directional migration cues that are a useful compromise between 2D and 3D environments, and allow us to characterize phenotypes that cannot be easily discerned in 3D environments and perform quantitative analysis on large cell numbers under defined conditions. Therefore, to first decipher how tumor architectures and mechanics influence T-cell motility, we designed "2.5D" nanotextured platforms with defined mechanical rigidities and oriented architectures. For soft to stiff "2.5D" platforms, we engineered high-definition nanotextured polyacrylamide gel (PAAG) surfaces with Hookean elasticity and shear moduli of $G' = 2.3, 8.6, 16, 50,$ and $>1000$ kPa[10,11], functionalized surfaces with either ICAM1 or ECM, and utilized these platforms to dissect architectural and mechanical regulation of T-cell phenotypic plasticity and cell migration.

Examination of T-cell behavior revealed that these platforms are sterically interactive, as they concurrently allow for both T-cell "in-groove" invasiveness (i.e., sterically interactive pseudopodia) into textures an "on-ridge" pseudopodia spreading behavior (see coexisting "on-ridge" and "in-groove" phenotypes in Fig. 1 and Supplementary Figs. 1–3). The "noninvasive" part of the T-cell nanotexture interface is structured more like a flat lamellipodium spread atop of the nanotextured plane (i.e., more lamellipodia-like mesenchymal behavior at the "on-ridge" level; Fig. 1a, Supplementary Figs. 1–3, and depicted in Fig. 1b). It is

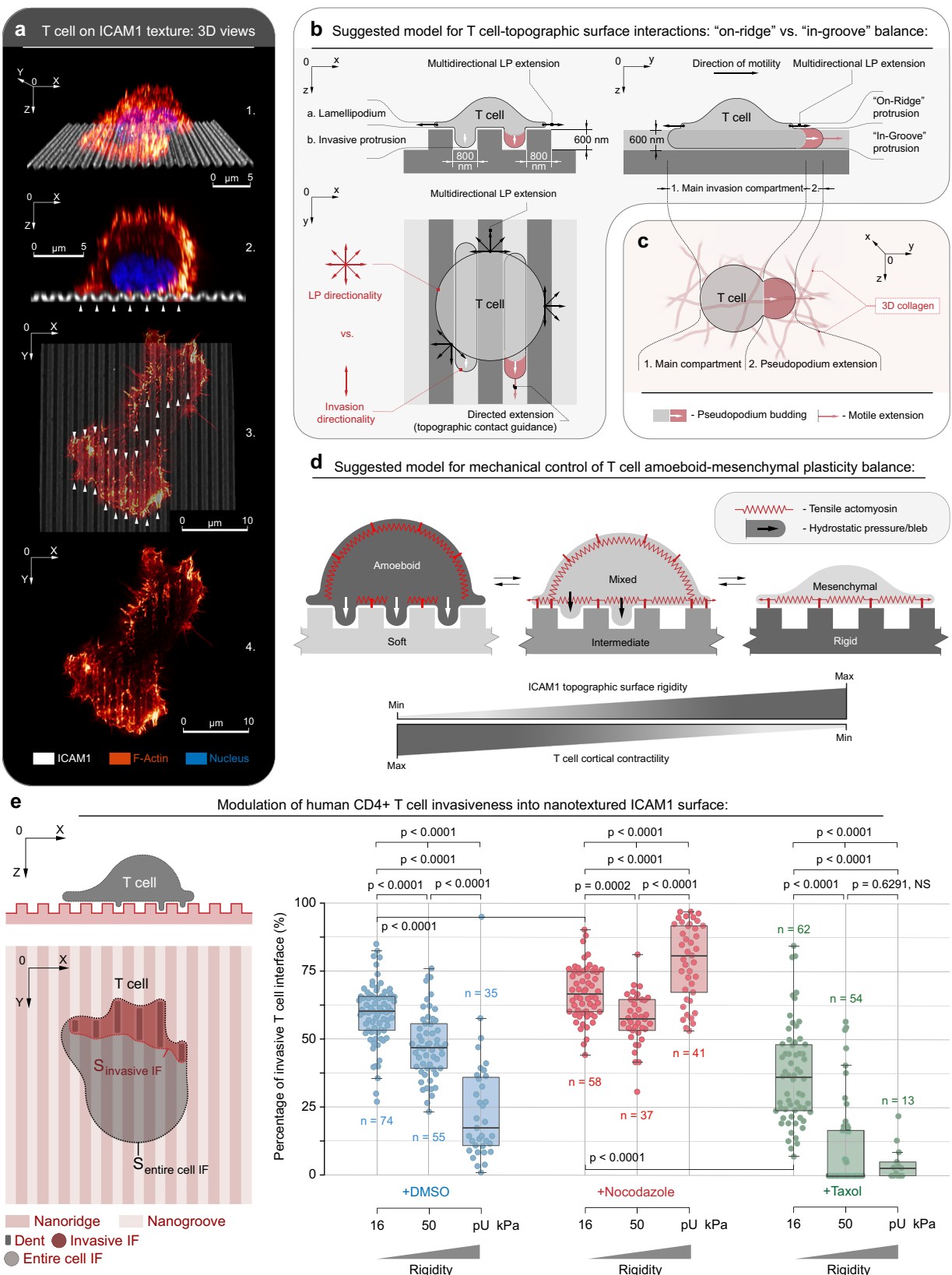

noteworthy that we define the mesenchymal-like phenotype for T cells as lamellipodium-driven flat adhesive spreading atop the nanoridges (Supplementary Fig. 3), similar to that observed during migration on flat environments[24] and consistent with electron microscopy findings of T-cell lamellipodia spread on top of very stiff nanostructured surfaces[29,30]. This is also consistent with carcinoma cell behavior where competitive dynamics between nanogrooves-guided directed invasive protrusions and less oriented "on-ridge" spreading can influence cell orientation and migration[11]. Furthermore, we note that in activated primary

**Fig. 1 Mechanical and microtubule regulated T-cell interactions with cancer-relevant nanoarchitectures. a** Super-resolution STED imaging used to generate 3D views of T cells on ICAM1-PAAG nanotopographies ($G' = 50$ kPa in this example) to characterize T-cell interactions with nanotopography. Views: 1, stereometric; 2, cross-section; 3, interface from atop (arrowheads, "in-groove" invasions); and 4, F-actin (phalloidin). Colors: white, ICAM1; fire, F-actin; blue, nuclei. **b** Projection schematics of nanotexture–T cell interactions, showing simultaneous "on-ridge" and "in-groove" dynamics. The nano-"ridge/groove" configuration supports both (1) unconfined, multidirectional "on-ridge" lamellipodium (LP), and (2) "in-groove" sterically interactive pseudopodium. "In-groove" T-cell dentations are sterically confined and are guided along the nanogrooves, driving T-cell contact guidance. **c** T-cell motility in 3D extracellular matrix where extension of invasive pseudopodium and rear retraction propel T cells through the 3D environment. Although 3D pseudopodia may not be confined to singular motility direction, they remain analogous to directed "in-groove" dentations. **d** Our model for the mechanomodulated "in-groove" (i.e., more amoeboid-like phenotype with actin-rich pseudopodia) and "on-ridge" (i.e., more mesenchymal-like phenotype, which for T cells is defined as lamellipodia driven protrusions during migration, particular on flat environments) dynamics can be competitive and regulate the amoeboid-mesenchymal plasticity balance. Our model predicts "on-ridge" mesenchymal spreading enhancement on rigid nanotopography outbalances and antagonizes "in-groove" amoeboid invasiveness. Alternatively, soft nanotextures shift the balance towards amoeboid "in-groove" steric cell invasiveness. **e** Metrics (left) and measurement (right) of T-cell "in-groove" relative invasiveness as a ratio between area of invasive regions and entire T-cell area at the topography interface (see Eq. 1 in the "Methods" section). Invasiveness is a function of nanotopography rigidity $G'$ and microtubule stability. $G'$ range: 16–50 kPa (PAAG) to ≫1000 kPa (pU: polyurethane plastic). Remarkably, at ICAM1-PAAG nanotextures of $G' < 16$ kPa hCD4+ T cells fail to attach to the nanotopographies, indicating $G' \approx 16$ kPa may be the lower mechanosensing limit for hCD4+ T cells on ICAM1. Individual dots correspond to the individual cells. Box plots depict the 25th percentile, median, and 75th percentile, and whiskers depict the 95% confidence intervals. Statistical tests are one-way ANOVA, Tukey's multiple comparisons tests. All n- and p-values are shown on the plots. Number of replicates per condition: 3. Source data are provided as a Source Data file.

T cells where we observe partial T-cell invasiveness into the nanogrooves depth of elastic nanotextures (i.e., the "in-groove" plane), we do not observe any filopodia precursors or flat protrusion along the groove walls, but instead observe a rounded phenotype (Fig. 1a and Supplementary Figs. 1 and 2), suggesting a more amoeboid-like behavior. Thus, in our model, we draw parallels between invasive "in-groove" pseudopodia (Fig. 1a) and T-cell pseudopodial protrusions observed during migration through 3D collagen matrices[12], and to the amoeboid locomotion reported within and along the uniaxially confining microchannels[25,26] (depicted in Fig. 1b, c). However, as the nanogroove platforms developed here facilitate partial interactions that allows for both highly directed "in-groove"-confined protrusions and less directed "on-ridge" 2D mesenchymal-like lamellipodium protrusions (Fig. 1a, b), i.e., a superposition of amoeboid and mesenchymal functional phenotypes (Fig. 1b), they allow us to define the mechanical regulators that drive the amoeboid-mesenchymal plasticity balance (Fig. 1d, e) and develop design criteria for us to tilt the phenotype balance, to favor more of one phenotype over the other, to ultimately be able to design customized T-cell motility phenotypes that can more effectively move throughout various types of TMEs.

From a mechanobiology perspective, the fact that we did not observe any filopodia or flat protrusion structures along the walls in nanogrooves of any stiffness, that invadopodia are very small and distinct punctate circular structures[31], and that lamellipodia require larger dimensions than the nanogrooves here provide and do not undergo sub-micron bending[32–34] suggests to us that "in-groove" amoeboid pseudopodia formation is primarily initiated by the cytoplasm hydrostatic pressure[35] generated by high cortical actomyosin contractility. Cytoplasm pressure-driven off-cortex plasma membrane peeling can result in budding of the plasma membrane blebs that can be pseudopodial precursors[36–38], similar to depictions in Fig. 1b, which then rebuild an actomyosin cortex in the new pseudopodia extensions[36–38]. Therefore, we hypothesized that in-groove pseudopodia formation dynamics can be inherently antagonized by mechanical tension from the cell–substrate adhesion interface from the "on-ridge" pseudopodia, i.e., the mesenchymal-like T-cell adhesion mechanisms (Fig. 1b, d), similar to observations in carcinoma cells[11]. This suggests that the mechanical properties of the substrate, and the related intracellular mechanics, would play a key role in governing directed motility of T cells. To explore this regulation, we first engineered nanotextures ("2.5D") and

nanolines (flat 2D lines) of different stiffnesses functionalized with ICAM1. Evaluation of activated human T-cell dynamics on "2.5D" substrates demonstrates mechanoresponsive behavior with significantly decreased "in-groove" interactions (i.e., "in-groove" invasiveness) as nanotexture stiffness increases (Fig. 1e), consistent with our hypothesis. In fact, increasing mechanical rigidity from $G' = 16$ kPa through 50 kPa (PAAG) to $G' \gg 1$ GPa (polyurethane, i.e., pU) results in a significant and consistent drop of "in-groove" invasiveness (defined as the invasive area percentage of cell–substrate interface projection) from ~60% to ~45%, to ~20%, respectively (Fig. 1e and Supplementary Fig. 1a–c), demonstrating that the degree of T-cell "in-groove" amoeboid-like invasiveness is regulated by variable mechano-transduction resulting from differences in the mechanical rigidity of the microenvironment. This observation suggests that as the substrate stiffness increases, the "on-ridge" lamellipodium-driven T cell-to-substrate adhesion interface strengthens via increased traction forces from the mechanosensitive adhesion signaling feedback (i.e., the mesenchymal-like portion of the cell consistent with reported T-cell mechanoresponsive behavior[24,39–41]). Thus, we conclude that the "on-ridge" mesenchymal adhesive cell–substrate interface competes with pseudopodial protrusions, i.e., invasive protrusion initiated by blebbing, which is driven by cytoplasm hydrostatic pressure from high cortical contractility. Indeed, our data suggest that the mechanical conditions of the T-cell environment can influence a redistribution of forces between high cortical contractility and on-ridge cell–substrate traction, and therefore manipulating this balance can influence T-cell phenotype and motility.

**MTs regulate the amoeboid-mesenchymal plasticity balance.** MT-rich protrusions provide a scaffold that sterically and mechanically enhances "in-groove" invasiveness into "2.5D" surfaces, to promote cell alignment and directed migration of carcinoma cells, i.e., contact guidance[11]. Furthermore, Nocoda-zole treatment has been shown to selectively increase T-cell blebbing and enhanced pseudopodia-like shape-shifting dynamics[42], which we link here to the increased T-cell invasive interactions with nanotexture contact-guidance cues. Therefore, we hypothesized that MTs may play a role in regulating T-cell interactions with nanoarchitectures and the degree of the balance between coexisting amoeboid and mesenchymal-like behaviors.. To test a role for MTs in regulating phenotype in human T cells, we pharmacologically destabilized MTs with Nocodazole or

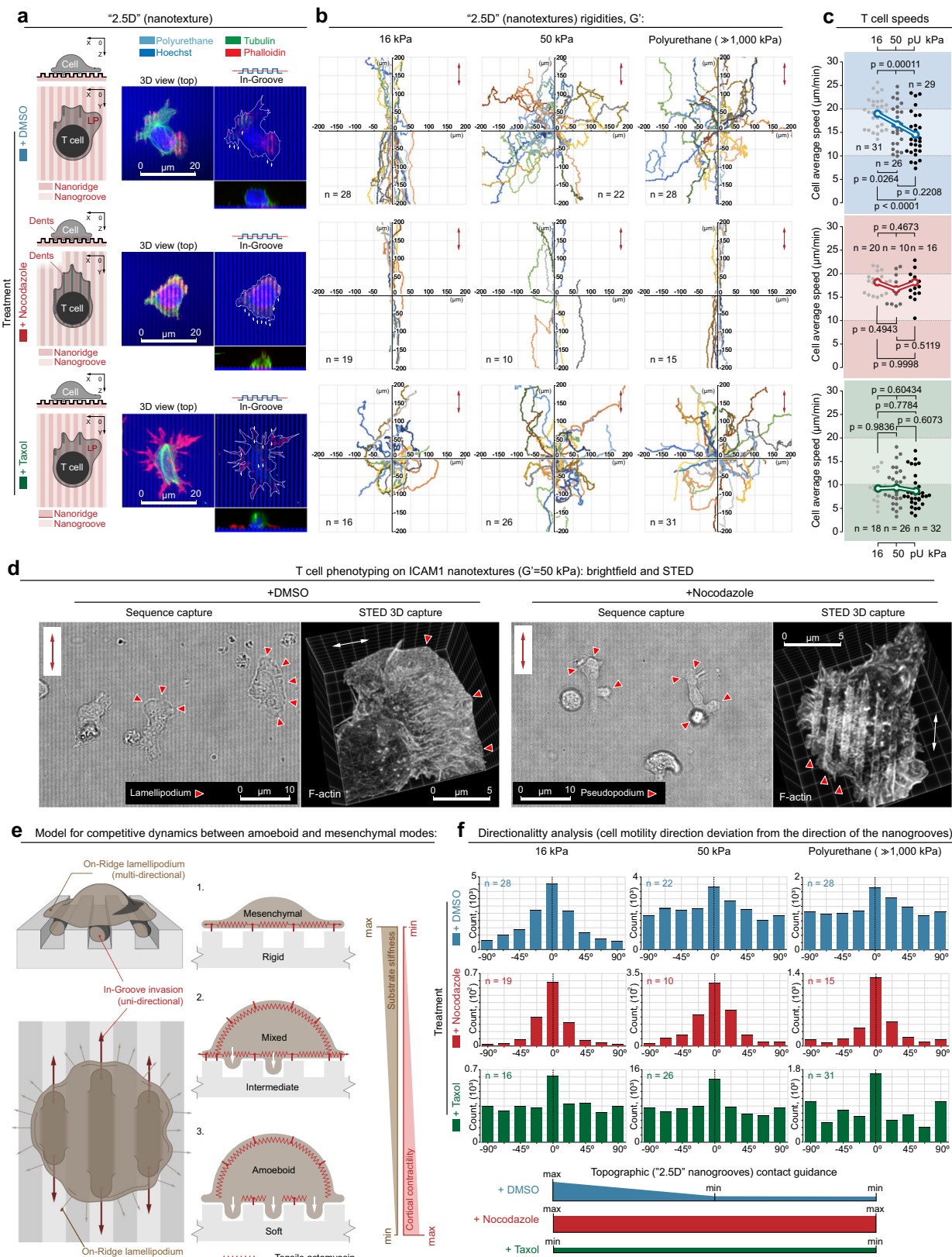

stabilized MTs with Taxol and evaluated resulting "in-groove" invasiveness (MT destabilization and stabilization are shown in Fig. 2a and Supplementary Fig. 1 and 2). Perhaps, surprisingly, in contrast to carcinoma cells, MT destabilization (+Nocodazole) profoundly increases T-cell amoeboid "in-groove" invasiveness (area fraction > 50%) across all nanotexture mechanical rigidities

(Fig. 1e and Supplementary Fig. 1), suggesting it pushes cells into a more robust cortical contraction state across all conditions. In stark contrast, MT stabilization with Taxol induced a universal and substantial reduction of "in-groove" invasiveness for each substrate stiffness (area fraction < 10–35%) without the loss of "on-ridge" T-cell adhesion and spreading across any of the "2.5D"

**Fig. 2 Microtubule dynamics regulate texture- and stiffness-dependent directed migration. a** Left to right: schematic, 3D micrograph, and "in-groove" cross-section and side views of human T cells on ICAM1 nanotextures (grooves/ridges widths = 800 nm). Top to bottom: control (+DMSO), destabilized MTs (+Nocodazole), and stabilized MTs (+Taxol) showing increased "in-groove" actin-rich pseudopodia in cells with destabilized MTs, in contrast to more mesenchymal-like behavior with Taxol. Nocodazole-induced MT disassembly and Taxol-induced MT stabilization are shown on immunofluorescence panels (green). Arrowheads indicate "in-groove" protrusions. Colors: light blue, polyurethane nanotexture; green, microtubules; red, F-actin; dark blue, nuclei. **b** T cells migration tracks on compliant ($G' = 16$ kPa), intermediate ($G' = 50$ kPa), or rigid ($G' \gg 1000$ kPa) ICAM1 nanotextures, where compliant nanotopographies enhance contact guidance. Top to bottom: T-cell migration under control (+DMSO), Nocodazole, or Taxol treatment conditions, where MT disassembly results in the enhanced directed migration across all rigidities. All *n*-values are shown on the plots. Number of replicates for per case: 3. Source data are provided as a Source Data file. **c** T-cell averaged per cell speeds on ICAM1 nanotextures of various rigidities (16, 50, and $\gg$1000 kPa) for control (+DMSO, top), Nocodazole (middle), and Taxol (bottom) conditions. Individual dots correspond to the individual cells. Statistical tests are one-way ANOVA, Tukey's multiple comparisons test. All *n*- and *p*-values are depicted on the plots. Number of replicates for each condition: 3. Source data are provided as a Source Data file. **d** T-cell phenotype transitions from the mixed "in-groove" and "on-ridge" phenotype to the more amoeboid "in-groove" dominant phenotype after MT disassembly from Nocodazole treatment. Micrographs are generated with human CD4+ T cells on rigid polyacrylamide gels ($G' = 50$ kPa), functionalized with human ICAM1. Sequence 1 capture (see Supplementary Movies 1 and 2) displays a well-developed lamellipodium and non-aligned motility during vehicle treatment as a control condition, whereas Nocodazole treatment induces amoeboid dynamics, pseudopodia, and migration along the nanogrooves (Sequence 2 capture; Supplementary Movies 3 and 4). Corresponding STED super-resolution images show detailed T cell–nanogrooves interactions. Arrowheads indicate lamellipodia in DMSO (left) or pseudopodia in Nocodazole (right). **e** Left, schematics of our model for T cell–ICAM1 nanotexture interactions. Right, mechanosensitive "in-groove" steric invasion and "on-ridge" mesenchymal-like (lamellipodial-based) interactions as a function of the mechanical rigidity of the microenvironment and cell contractility. **f** Quantification of directionality of human T cells migration during contact guidance as a function of substrate mechanical rigidity and the state of microtubules. Corresponding T-cell migration tracks are shown in the same matrix order in **b**. Measurements represent frequency distributions of cell-to-nanogroove angles every 10 s step. All *n*-values are indicated on the plot. Number of replicates for each condition: 3. Source data are provided as a Source Data file.

substrate rigidities (Fig. 1e and Supplementary Fig. 1). Hence, altering MT abundance or dynamics changes the fundamental interactions between T cells and their environment, and thus the T-cell phenotype (Supplementary Fig. 3). This suggests that MT state may influence T-cell migration, which can inform strategies to enhance T-cell migration in tumors, and also suggests that clinically utilized drugs such as Taxol could be detrimental to T-cell migration.

**MTs regulate texture- and stiffness-dependent migration.** To define the impact of stiffness on directed T-cell migration, we analyzed cell directionality relative to nanotopography (i.e., contact guidance) and migration speed on aligned substrates with stiffnesses ranging from 16 kPa to over 1 GPa (Fig. 2a–c), a range where we observed super-positioned phenotypes in response to the "2.5D" nanoarchitectures (Figs. 1 and 2d, and Supplementary Figs. 1–3). By tracking cell movement over time, we identified distinct differences in T-cell contact guidance between soft and rigid conditions. On lower modulus nanotexture conditions, T-cell migration is strongly contact guidance-driven with highly oriented migration along the direction of the nanogrooves (Fig. 2b). In contrast, cells on stiffer substrates display more random migration tracks relative to the directionality of the underlying nanogrooves (Fig. 2b). Notably, this observation again indicates that "in-groove" invasive amoeboid-like pseudopodia provide stronger steric interactions between T cells and the nanotexture, which are abundantly observed in T cells on soft nanotextures (Fig. 1e). Alternatively, mechanosensitive tension enhancement at the T-cell interface with rigid nanotextures (mesenchymal-like behavior) favors the flattened noninvasive "on-ridge" interface with reduced steric interactions with the anisotropic nanogrooves, i.e., a lower contact-guidance response. Thus, this demonstrates that the shift toward a more amoeboid phenotype regulates migration in structurally complex environments in a substrate stiffness-dependent manner (Fig. 2e). Indeed, quantification of T-cell migration directionalities per every 10 s-step across distinct nanotexture rigidities (16–50 kPa to >1000 kPa) demonstrates significant differences in T-cell migration directionality, with collectively stronger guidance along the 0° angle of deviation from the nanogrooves orientation on soft ($G'$

= 16 kPa) nanotextures, but more random and less persistent directionalities on stiffer nanotextures (Fig. 2f). That is, more than 75% of the T-cell population migrate within only ±20° of deviation from the nanogrooves axis on the soft ($G' = 16$ kPa) nanotextures at any given moment of time. On the contrary, all stiffer nanotextures ($G' = 50$ kPa or greater) induce only ~50% of T-cell population migrating within ±40° of deviation range from the axis of the nanogrooves. Thus, the softer nanotextures preset T cells to more robustly sense and conform their migration dynamics to the topography of their microenvironments.

Moreover, as expected from our analysis of T-cell "in-groove" vs. "on-ridge" interactions with nanotexture (Fig. 1), in control conditions (+DMSO), T-cell migration speed displays a direct and proportional correlation to the degree of T-cell "in-groove" invasiveness (i.e., the amoeboid-mesenchymal phenotype balance), with significantly slower migration as substrate stiffness increases (i.e., shifted toward a larger contribution from the "on-groove" mesenchymal phase; Fig. 2a–c, +DMSO). This again suggests that softer environments result in higher T-cell cortical contractility that more robustly facilitates fast directed migration. That is, softer environments result in weaker T cell-to-nanotexture "on-ridge" adhesion and spreading, allowing for mechanical dominance of T-cell cortical contractility that promotes "in-groove" amoeboid dynamics and more robustly facilitates fast directed migration. Furthermore, importantly, the principles of the mechanically regulated amoeboid-mesenchymal balance that we report here and its effects on T-cell-directed migration are not limited to interactions with ICAM1, as we observe similar trends for T-cell migration along fibronectin (FN) nanotextures of the same mechanical rigidity range (Supplementary Fig. 4). Again, directionality and speed are maximal on softer FN nanotextures (Supplementary Fig. 4), suggesting that the architecture and mechanics of the environment are dominant factors that can drive T-cell motility across a range of complex ECM environments. These cumulative data also suggest that the increased tissue stiffness associated with TMEs (vs. normal tissues) may in fact reduce T-cell migration, in contrast to behavior on flat substrates[24], and be part of the physical barrier to effective T-cell sampling in tumors. In fact, on flat nanolines we observe distinct phenotype-dependent migration speed (Supplementary Fig. 5). Furthermore, we further tested our hypothesis

using inhibition of Formins, where inhibition does not substantially inhibit lamellipodia spreading but are needed for force transmission between integrin adhesions in the lamella and the substrate[43–47], and observed a profound shift in the phenotype balance toward a more "in-groove" T-cell phenotype (Supplementary Fig. 6). On stiff substrates, this results in significantly greater directed migration (Supplementary Fig. 6), which is consistent with our hypothesis that forces from the mesenchymal-like "on-ridge" level compete with cortical contractile forces that drive the phenotype balance more toward the amoeboid phenotype. Thus, we conclude that "in-groove" invasiveness resulting from high cortical contractility and confined pseudopodia promotes more efficient directed migration. Therefore, we suggest a mechanistic model for efficient T-cell-directed migration where "in-groove" invasive pseudopodial protrusions sterically guide and drive T-cell migration along the nanogrooves (depicted in Fig. 2e), similar to the established amoeboid-like leader bleb-guided cell migration scenario[48], but with antagonizing "on-ridge" mesenchymal lamellipodium. Thus, this provides the basic principles governing amoeboid-mesenchymal plasticity control and some of the design rules to produce more efficient migration in mechanically complex environments.

As destabilization of MTs with Nocodazole induces a strong T-cell phenotype shift towards amoeboid "in-groove" invasiveness across all "2.5D" substrate rigidities (Fig. 1e and Supplementary Fig. 1), we hypothesized that disruption of MTs would enhance T-cell migration across a range of mechanical environments. Indeed, MT disruption results in a shift towards more "in-groove" protrusions (i.e., tilts to the phenotype balance toward a more amoeboid phenotype; Fig. 2d, Supplementary Figs. 1–3, and Supplementary Movies 1–4) and highly directed migration from contact guidance across all substrate stiffnesses on both ICAM1 and FN (Fig. 2a, b and Supplementary Fig. 4a, b). Likewise, under MT destabilization, cell speed remains consistently high regardless of substrate stiffness (Fig. 2a, b and Supplementary Fig. 4a, b), suggesting a shift toward high cortical contractility regardless of substrate stiffness, and that following the shift toward the more amoeboid phenotype, T cells are now able to migrate as efficiently in stiff environments as in soft environments. Interestingly, MT disruption also impacts the response to flat nanolines by severely limiting development of mesenchymal behavior, while still resulting in no change in speed in response to increasing stiffness, albeit with a lower overall speed than observed with nanotextures (Supplementary Fig. 5a–c), consistent with our hypothesis. In stark contrast to enhanced migration following MT destabilization, MT stabilization with Taxol induced the opposite trend. Taxol-treated T cells showed a substantial reduction of "in-groove" invasiveness (area fraction < 10–35%) and significant decreases in directionality and speed (Fig. 2a–c), indicating a shift from the amoeboid phenotype toward the more mesenchymal "on-ridge" T-cell spreading (Supplementary Fig. 1), which is also supported by findings on flat 2D substrates, where Taxol-treated cells shift toward the more mesenchymal phenotype (Supplementary Fig. 5). Thus, our results indicate that commonly utilized MT-stabilizing taxane chemotherapeutics may significantly reduce T-cell migration, suggesting that analyzing migration of MT-stabilized T cells in tumor environments is needed, as caution may be warranted when timing chemotherapy and immunotherapy combination approaches that require T-cell movement through tumors. However, our data indicate that more destabilized MTs promote T-cell migration in architecturally and mechanically relevant "2.5D" environments, and offer insight into design criteria for approaches to enhance T-cell migration in more complex 3D environments using pharmacological agents or via genome engineering. Thus, our data suggest

that this approach could ultimately be applied to further improve therapeutic T cells by enhancing migration in cells that are already engineered with high-affinity tumor-reactive T-Cell Receptors (TCRs) specific to tumor antigens.

**Controlling 3D migration through the MT–contractility axis.** As our nanotextured platforms were designed to capture key alignments and textures observed in tumor collagen fiber networks[6,8,11], and our data suggests parallels between invasive "in-groove" pseudopodia and T-cell pseudopodial protrusions observed during migration through 3D collagen matrices[12], we next sought to test the hypothesis that disrupting MTs can enhance 3D motility. Three-dimensional fibrous ECM networks with more random ECM organization (Fig. 3a) pose a greater challenge to effective migration, as cells move as a series of short contact-guidance steps along fibers, or groups of fibers, with the need to re-orient in order to transition to another fiber configuration for guidance cues or through tortuous porous paths. In this context, from our data on "2.5D" substrates and reports highlighting increased Rho-mediated contractility via GEF-H1 following MT destabilization[42,49–52], we hypothesized that MT destabilization will promote 3D migration and is influencing increased Rho activity behavior via GEF-H1 (Fig. 3b), to produce robust cortical contractility that drives bleb-to-pseudopod protrusions (Fig. 3c). Indeed, we attribute our observed phenotype under MT-destabilized condition to the release of MT-sequestered GEF-H1, which activates the RhoA pathway that directly enhances downstream actomyosin contractility[49–52]. We first tested our hypothesis by pharmacologic MT destabilization (+Nocodazole), perturbing RhoA activity, and disrupting downstream actomyosin contractility.

Using atomic force microscopy (AFM)[53] on primary human T cells, we tested our conclusions by manipulating MT stability and the overall MT–contractility axis with various pharmacologic agents to alter MT stability, Rho contractility, and myosin motor activity (Fig. 3d). With our AFM protocol, the corresponding actomyosin cortical tension contractile forces are balanced by opposite forces generated by the intracellular "cytosolic" hydrostatic pressure[53–55] (Supplementary Fig. 7). Importantly, the cellular cortical tension and the accompanying intracellular hydrostatic pressure concomitantly regulate the generation of cellular blebs and amoeboid migration in complex 3D microenvironments[36,56]. Consistent with our model, results indicate that MT destabilization significantly increases Rho activity ~2.5-fold and the mechanical rigidity as a function of robust cortical actomyosin contractility (Fig. 3 and Supplementary Fig. 7), which is associated with the amoeboid phenotype and confirmed by treating cells with Blebbistatin to robustly disrupt myosin-regulated contractility (Fig. 3d, +DMSO vs. +Nocodazole). In fact, we note the highly consistent T-cell softening without myosin-based contractile forces to the nearly identical values observed during treatment with +Blebbistatin vs. +Nocodazole + Blebbistatin (Fig. 3d). Furthermore, consistent with our cortical tension observations, MT destabilization significantly increases the intracellular hydrostatic pressure, whereas Blebbistatin treatment reduces hydrostatic pressure (Supplementary Fig. 7). Interestingly, Taxol also induces T-cell rigidification and increased hydrostatic pressure; however, this stiffening was more independent of actomyosin contractility, as addition of Blebbistatin did not soften cells or reduce pressure as profoundly as in control or Nocodazole treatment conditions (Fig. 3d and Supplementary Fig. 7). Thus, we attribute this significant effect to rigidification via Taxol-stabilized MTs[57–59] and may also be related to findings that the MT cytoskeleton can physically connect and interact with the actomyosin cortex providing an additional structural scaffold[60,61],

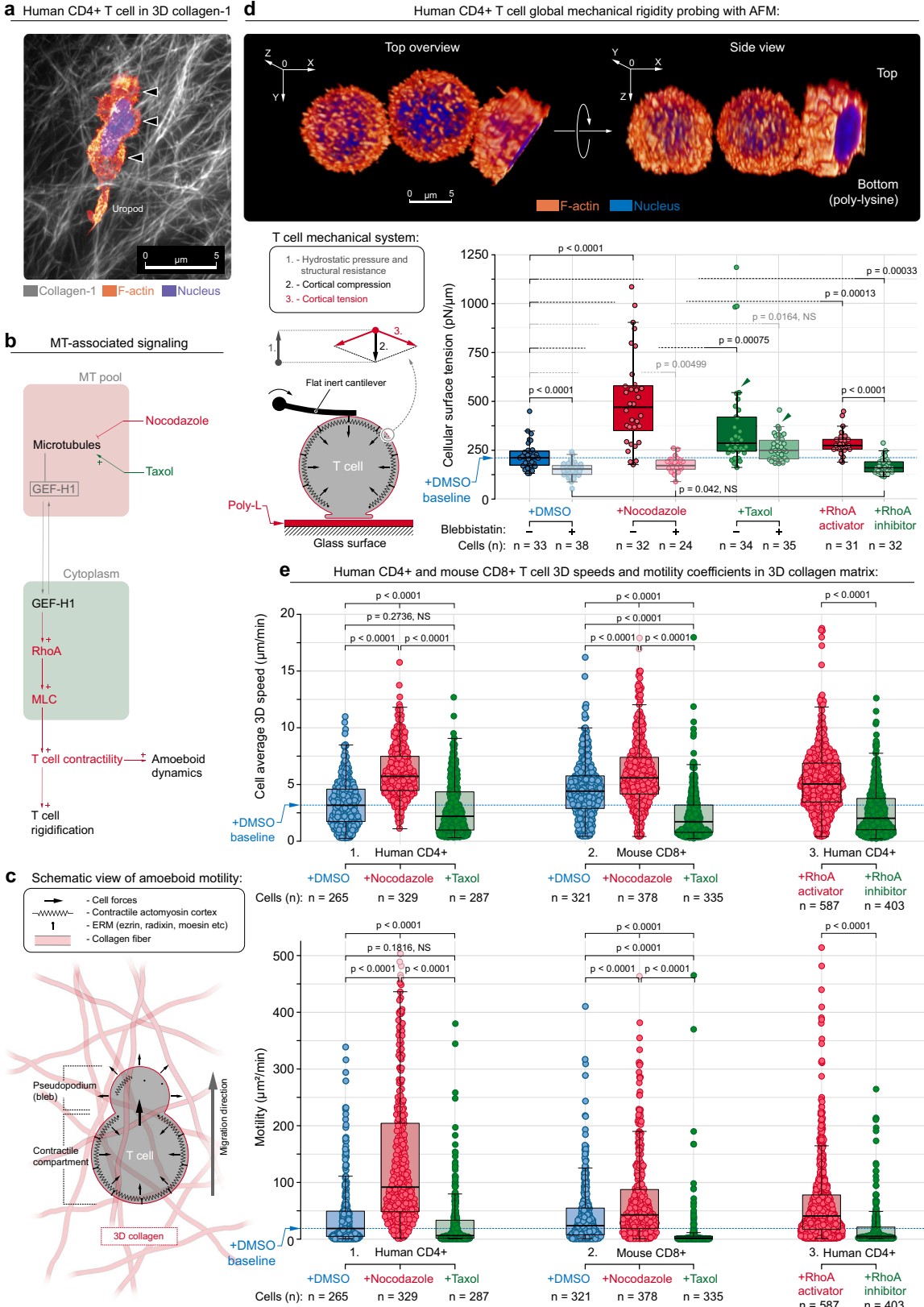

which is also strongly in line with the recently identified contribution of scaffolding MTs in active cell mechanics during directed migration[11]. Lastly, direct RhoA activation was performed with G-switch treatments, which activates Rho by converting glutamine-63 to glutamate in the Switch II region, in order to constitutively activate Rho, whereas inhibition was performed with cell-permeable C3 exoenzyme. Again, consistent with our hypothesis, activation of Rho increases T-cell rigidity and hydrostatic pressure, whereas Rho inhibition softens cells and reduces hydrostatic pressure to levels observed following treatment with Blebbistatin (Fig. 3d and Supplementary Fig. 7). Collectively, this suggests that amoeboid-mesenchymal plasticity balance is indeed

**Fig. 3 The microtubule–contractility axis regulates T-cell mechanics and migration in 3D environments. a** STED micrograph of a human T cell migrating through 3D collagen matrix. Arrowheads, distinct bleb-to-pseudopodia amoeboid protrusions. Colors: fire, F-actin; white, collagen fibers; blue, nuclei. **b** Model for how MT-associated signaling integrates amoeboid motility through MT stability and contractility via GEF-H1 cytoplasm⇌MT reversible transitioning. **c** In our model, T-cell amoeboid motility in 3D environments is linked to actomyosin contractility where cortex tension-induced hydrostatic pressure forces plasma membrane off-cortex peeling (blebbing) that results in pseudopodia formation. Continuous pseudopodial formation facilitates steric interactions between T cells and the 3D microenvironment, i.e., persistent amoeboid motility. **d** Top, 3D reconstruction of the probed T cells in AFM testing settings. T cells non-specifically adhere to the poly-lysine-coated plasma-treated glass surface (bottom), whereas the AFM cantilever mechanically tests cells from atop. Bottom left, mechanical testing of T cells with atomic force microscopy (AFM) to measure the global compression through a flat cantilever, to measure the overall cortical tension that is counterbalanced with cytoplasm hydrostatic pressure and additional intracellular cytoskeletal structural resistance (microtubules, nucleus, etc). Colors: fire, F-actin; blue, nuclei. Bottom right, AFM measurement of global T-cell surface tension, a measure of cortical contractility. The control group and each of the two MT-targeting treatments (i.e., +DMSO, +Nocodazole, and +Taxol, solid colors) are paired with blebbistatin co-treatment (pale semi-transparent colors), to verify the key role of actomyosin contractility as a regulator of changes in cell rigidity during MT targeting. Alternatively, direct RhoA activation or inhibition is compared to MT perturbation results. Both MT disassembly (+Nocodazole) and direct RhoA activation (+RhoA activator) induce mechanical rigidification of human CD4+ T cells via increased actomyosin tension, as demonstrated by AFM findings after blebbistatin co-treatment. It is noteworthy that MT destabilization or direct RhoA activation induce a rise of hydrostatic pressure. Also, Taxol-induced MT stabilization increases passive (i.e., actomyosin-independent) T-cell rigidity via a direct mechanical contribution from stabilized MT scaffolds (+Taxol vs. +Taxol + Blebbistatin treatments; arrowheads). Individual dots correspond to individual cells. Box plots depict the 25th percentile, median, and 75th percentile, and whiskers depict the 95% confidence intervals. Statistical tests are pairwise one-sided *t*-tests. All *n*- and *p*-values are shown on the plots. Number of replicates per condition: 3. Source data are provided as a Source Data file. **e** 3D migration speed and overall 3D cell motility of T cells in 3D collagen-FN matrix: 1, human T cells (hCD4+) in the presence of vehicle (+DMSO), Nocodazole (+Nocodazole), or Taxol (+Taxol); 2, mouse cytotoxic T cells (mCD8+) in the presence of vehicle (+DMSO), Nocodazole (+Nocodazole), or Taxol (+Taxol); and 3, human T cells in the presence of RhoA activator (+RhoA activator) or inhibitor (+RhoA inhibitor). Individual dots correspond to the cell motility from time-lapse imaging every 1.5 min for >1 h. Box plots depict the 25th percentile, median, 75th percentile, and whiskers depict the 95% confidence intervals. Statistical tests are one-way ANOVA, post hoc Tukey's tests. All *n*- and *p*-values are shown on the plots. Number of replicates per condition: 3. Source data are provided as a Source Data file.

regulated by a MT–contractility axis, and that, together with cortical tension, modulation of the intracellular hydrostatic pressure allows T cells to alter their shape, control their locomotion, and govern their mechanics, while infiltrating through complex intratumoral microenvironments.

Equipped with this new understanding of T-cell mechanics and their implications on T-cell motility, we sought to evaluate and enhance T-cell migration in more complicated 3D environments. Using 3D collagen-FN matrices, we first quantified T-cell speed and overall motility using the persistent random walk model (PRWM), for both primary human CD4+ (hCD4+) T cells that were available from healthy human donors (see "Methods") and mouse CD8+ (mCD8+) T cells where cytotoxic T cells from tumor-bearing hosts could be obtained from genetically engineered *KPC* mice bearing pancreatic tumors, which are highly faithful to the human disease[4,15,62]. In both human and mouse T cells, MT destabilization with Nocodazole significantly increases speed by ~50–100% and overall motility ~2- to 4-fold (Fig. 3e), again suggesting that MT-destabilizing agents have the potential to enhance T-cell migration in tumor-like architectures. Likewise, similar strategies may be employed to enhance Rho activation, which increases migration while inhibition of Rho again decreases speed and overall motility (Fig. 3e). Moreover, to further test this hypothesis, we isolated mCD8+ T cells from *KPC* mice harboring autochthonous metastatic pancreas cancer and imaged their migration over hours in paired live tumor slices (Fig. 4a), with or without MT destabilization. Consistent with data on "2.5D" and in engineered 3D environments, MT disruption, specifically in T cells and not other tumor cells, significantly increased migration of cytotoxic T cells through native tumor environments (Fig. 4b). In stark contrast, T-cell exposure (and not other tumor cells) to the MT-stabilizing agent Taxol results in a significant decrease in both speed and overall motility, both in 3D matrices in vitro (Fig. 3e) and in live tumors (Fig. 4b). This suggests that although Taxane agents can have a beneficial therapeutic impact on carcinoma cells, they may also limit the impact of native antitumor immunity, and that combining T-cell-centric immunotherapy approaches (e.g., checkpoint blockade) with taxane agents requires additional

scrutiny, as these chemotherapies may in fact limit cytotoxic T-cell migration into and through tumors. On the other hand, MT disruption can diminish carcinoma cell-directed motility[10,11] while promoting T-cell migration and therefore manipulating the MT-associated signaling and/or contractility has the potential to significantly enhance T-cell migration through ECM architectures found in tumors. Therefore, this may provide a strategy that could be developed to enhance migration of therapeutic T cells (e.g., CAR-T cells or other engineered T cells) or treat native T cells during immunotherapy (e.g., immune checkpoint blockade).

To further confirm our findings that the MT–contractility axis is a fundamental regulator of T-cell migration in complex environments and specifically test our hypothesis that GEF-H1 is a key mediator of MT to Rho-mediated contractility signaling, we employed CRISPR technology to knock out *GEF-H1* in three independent primary human T-cell lines (Supplementary Fig. 8) and analyzed their 3D migration (Fig. 5). To test our hypothesis that Rho activation from Nocodazole-induced MT instability is regulated by GEF-H1, we measured activated Rho levels and compared 3D migration of control Cas9 cells and *ARHGEF2 (GEF-H1)* KO cells in the presence of Nocodazole (Fig. 5a). Consistent with our hypothesis, we observed an ~2-fold increase in active Rho following Nocodazole treatment, which is abolished in *GEF-H1* KO cells (Supplementary Fig. 7) and that, under MT instability, migration is significantly decreased in *GEF-H1* KO cells compared to control cells expressing Cas9, with overall motility cut in half (Fig. 5a–c). However, remarkably, in separate experiments we discovered that GEF-H1 loss without MT destabilization significantly promotes migration (Fig. 5d–f). In three primary activated human T-cell lines, migration speed through 3D matrices is significantly increased (Fig. 5e), whereas the overall motility is significantly increased by more than 50% (Fig. 5f), showing that genome engineering to eliminate *GEF-H1* can result in engineered T cells that are more capable of migration throughout mechanically and structurally complex environments. Thus, this data demonstrates that engineering approaches to alter T-cell cytoskeletal-to-contractile machinery and signaling have the potential to significantly enhance T-cell movement through

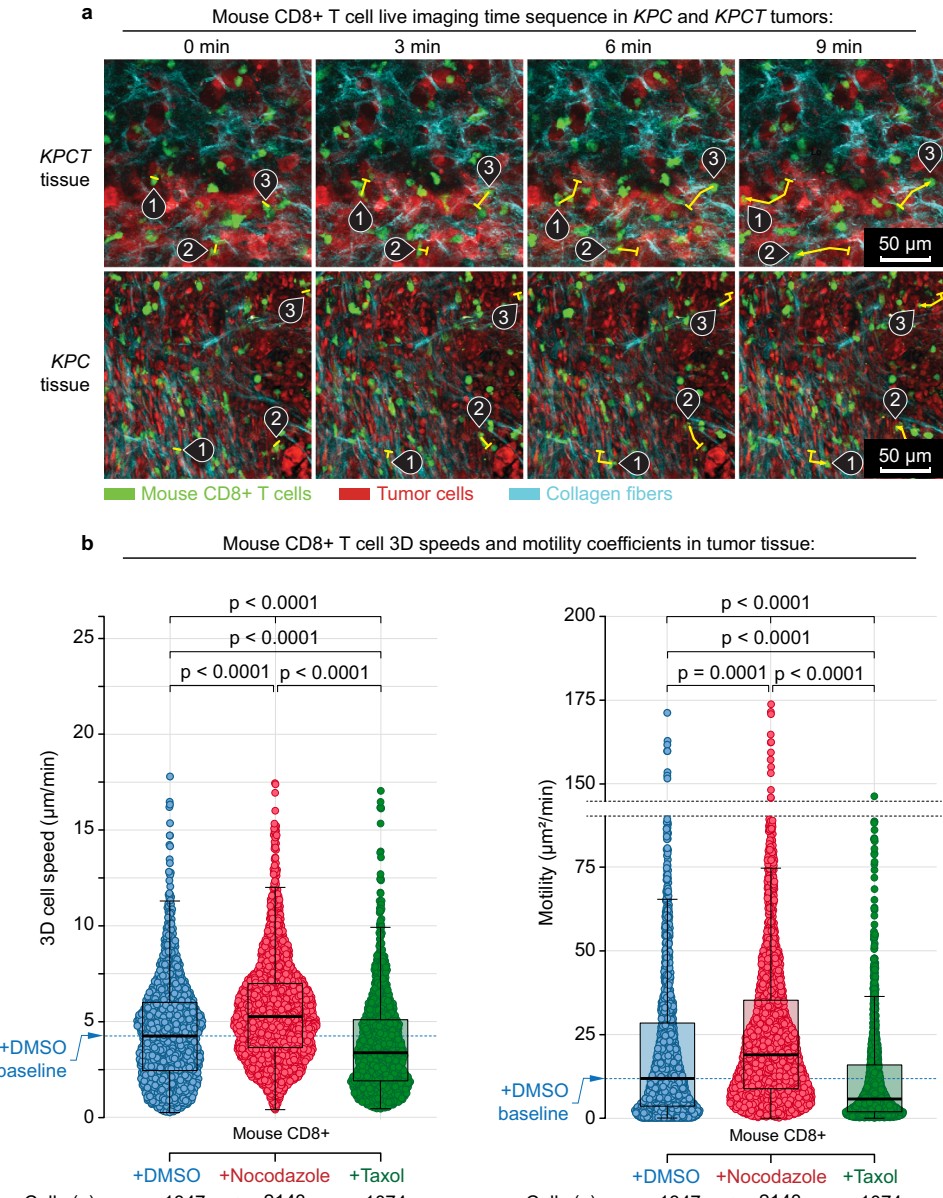

**Fig. 4 The microtubule–contractility axis regulates intratumoral migration of cytotoxic T cells. a** Combined multi-photon excitation of fluorescence (MPE) and second harmonic generation (SHG) of primary mouse CD8+ cytotoxic T cells (green) in paired live pancreatic tumor from *KPC* mice either expressing tdTomato specifically in carcinoma cells (*KPCT*; top row) or with all cells fluorescently labeled (*KPC*; bottom row). Sample migration tracks of individual T cells are highlighted using numerated pins along with their tracks (yellow curves: start position, tick; end position, arrowhead). Images are samples from imaging every 3 min for >1 h. Colors: green, T cells; red, tumor cells; cyan, collagen fibers. **b** Speeds (left) and motility coefficients (right) for cytotoxic T cells migrating within 3D tumor microenvironments under control (+DMSO), MT-destabilizing (+Nocodazole), and MT-stabilized (+Taxol) treatment conditions, where MT destabilization leads to hypercontractile states and increased migration through stroma dense native pancreatic adenocarcinomas, whereas treatment with Taxol significantly decreases the ability of T cells to migration through native tumor microenvironments. Individual dots correspond to the cell motility from time-lapse imaging every 1.5 min for >1 h. Box plots depict the 25th percentile, median, and 75th percentile, and whiskers depict the 95% confidence intervals. Statistical tests are one-way ANOVA, post hoc Tukey's tests. All *n*- and *p*-values are shown on the plots. Number of replicates per condition: 3. Source data are provided as a Source Data file.

complex solid tumor environments. Indeed, genome-engineering strategies, or focused pharmalogic targeting, to alter the MT–contractility axis are likely to enhance T-cell migration in the therapeutic setting where T cells are often not able to move throughout and sample an entire solid tumor volume.

## Discussion
The physical and molecular mechanistic principles that regulate the dynamic interactions between migrating T cells and their 3D

environments, e.g., the basis of T-cell 3D locomotion throughout normal tissues and solid tumors, remain elusive, largely due to the overall complexity of 3D environments and undeciphered phenotypic plasticity, such as coexisting amoeboid and mesenchymal modes of motility. Despite the widely acknowledged fact that the steric and structural density, architecture, and mechanics of solid tumors can prevent effective T-cell infiltration or distribution[1–5], the study of mechano- and structure-sensitive aspects of T-cell motility has predominantly focused on flat 2D environments (i.e.,

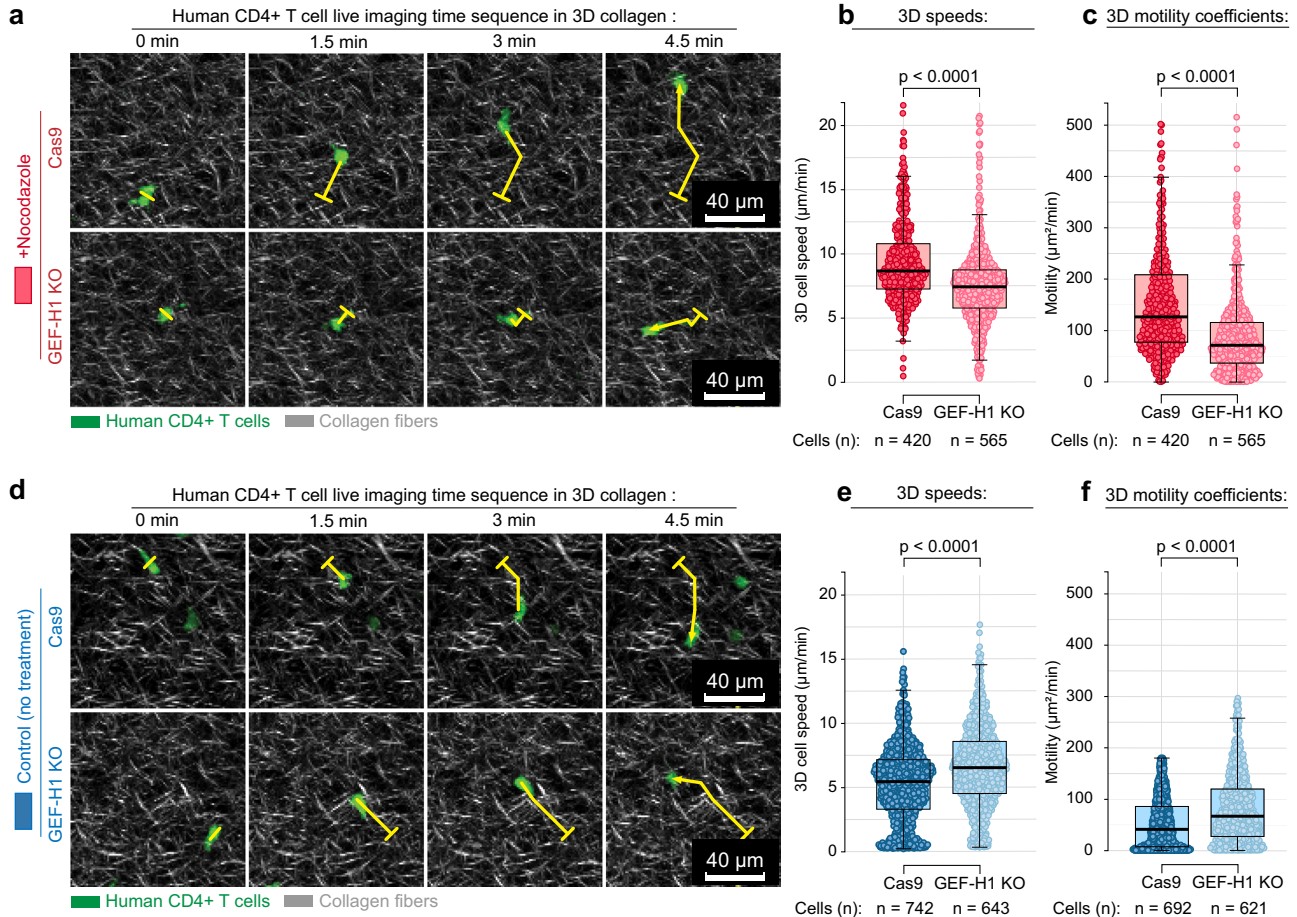

**Fig. 5 CRISPR *ARHGEF2/GEF-H1* knockout influences T-cell migration.** Although *GEF-H1 knockout (KO)* prevents enhanced migration from microtubule destabilization, it enhances migration of untreated T cells. **a** Combined multi-photon excitation of fluorescence (MPE) and second harmonic generation (SHG) of primary human T cells (green) migrating through 3D matrices. To test the hypothesis that increased contractility from destabilized microtubules is GEF-H1 dependent, we knocked out *GEF-H1* using CRISPR and tested migration over time in the presence of destabilized MTs (+Nocodazole). Samples of migrating T cells are shown and tracks of individual cells are highlighted with their tracks (yellow curves: start position, tick; end position, arrowhead). Images are samples from imaging every 1.5 min for >1 h. **b** Speeds (left) and **c** motility coefficients (right) for control (Cas9) and *GEF-H1* KO T cells migrating within 3D microenvironments under MT-destabilizing conditions (+Nocodazole) showing that loss of GEF-H1 disrupts the contractility-enhanced migration from MT destabilization. Individual dots correspond to the cell motility from time-lapse imaging every 1.5 min for >1 h. Box plots depict the 25th percentile, median, and 75th percentile, and whiskers depict the 95% confidence intervals. Statistical tests are pairwise one-sided *t*-tests. All *n*- and *p*-values are shown on the plots. Number of replicates per condition: 3. Source data are provided as a Source Data file. **d** Combined MPE/SHG of control and *GEF-H1* KO cells without destabilization of MTs. Samples of migrating T cells are shown and tracks of individual cells are highlighted using numerated pins along with their tracks (yellow curves: start position, tick; end position, arrowhead). **e** Speeds (left) and **f** motility coefficients (right) for control (Cas9) and *GEF-H1* KO T cells migrating within 3D microenvironments with unperturbed MTs showing increased migration through complex 3D environments following genome engineering to KO *GEF-H1*. Individual dots correspond to the cell motility from time-lapse imaging every 1.5 min for >1 h. Box plots depict the 25th percentile, median, and 75th percentile, and whiskers depict the 95% confidence intervals. Statistical tests are pairwise one-sided *t*-tests. All *n*- and *p*-values are shown on the plots. Number of replicates per condition: 3. Source data are provided as a Source Data file.

more mesenchymal-like phenotypes) or confinement in microchannels[23,24,26,27], which poorly mimic the structural and steric complexity of tumorous tissues. In this study, we attempted to comprehensively interlink the mechanistic and mechanosensing principles of 2D and 3D T-cell motility. As a result, we principally advanced the concept of super-positioned amoeboid-mesenchymal-like phenotypes in T cells and deciphered a mechanosensing role during the dynamically adaptive balance between amoeboid-mesenchymal phenotypes that impacts 3D migration. By focusing on key controlling factors, such as microenvironment architecture, mechanics, and T-cell myosin-driven contractility, we established that 3D amoeboid migration is proportional to cortical actomyosin contractility and tilting the

balance toward this phenotype provides a significant advantage for greater T-cell motility in both "2.5D" and 3D environments. Importantly, the increase in T-cell amoeboid behavior is inherently accompanied by increased nanoscale steric interactions with the ECM texture, indicating the leading role of T-cell shape shifting, with directed pseudopodia, as a driver of T cells shifting to a more amoeboid locomotion. Furthermore, increasing T-cell contractility either via direct RhoA activation or via the MT-GEF-H1 axis robustly drives up cortical contractility and desensitizes T cells to microenvironment mechanics, shifting T cells more toward an amoeboid mode, resulting in increased motility in dense 3D collagen-FN matrices and in *KPC* tumors. Thus, we define the starting basis for a new generation of T-cell engineering

strategies that address and tackle the issue of poor efficiency of T-lymphocyte movement throughout architecturally and mechanically complicated and heterogeneous microenvironments.

Stromal cell populations and the ECM surrounding cancerous cells in the pancreas are believed to be critically involved in tumor growth, metastasis, and resistance to therapy[15,63], and also limit effective antitumor immunity from T cells due to immunosuppression and fibrotic barriers[2,4,5,13,16–18]. Yet, following specific strategies to alter the stroma, PDA becomes susceptible to immune checkpoint blockade[17,18] and engineered T cells can overcome some of the physical and chemical barriers[64]. Thus, therapies that rely on T-cell function can be effective against PDA and other solid tumors, in the right setting. Yet, one key obstacle to maximally effective T-cell therapies in solid tumors is their inability to efficiently navigate through heterogeneous TMEs to sample the entire tumor volume within a functional time domain. Here we present an advance to overcoming this obstacle and show that perturbing the MT–contractility axis, and in particular activation of Rho-mediated contractility, results in T cells with increased capacity to move through solid TMEs. However, we note that although perturbing these elements certainly can improve T-cell migration, care must be taken when engineering cells to migrate better, to ensure they retain the ability for adequate activation. For instance, a number of studies have shown that MTs are not required for the immune synapse TCR formation, whereas other studies have suggested MT-dependent synapse behavior, where this may also depend on the type of MT-targeting drug, e.g., examples of immune synapse disruption under nocodazole but not vinblastine[65–68]. Thus, there likely may be ways to perturb MT stability or MT-associated signaling without loss of immune synapse function; however, either way, perturbing the contractility side of the axis, and likely other adhesion-to-motor elements as well, alleviates potential issues with altered MT dynamics and clearly shows great promise for enhancing T-cell migration in tumors. Thus, specific strategies, such as pharmacologic interventions and genome engineering, to enhance contractility either through elements of adhesion-to-cytoskeleton-to-Rho signaling pathways or direct alteration of contractility-governing molecular motors have a role in T-cell engineering to generate "mechanically optimized" therapeutic cells. This first volley to overcome barriers in the TME provides a foundation from which to build upon. With the recent advances in genome engineering, such as the use of CRISPR technology here, we are well poised to employ our initial design criteria for future studies to test rational engineering strategies, to improve the ability of T cells to move throughout, and sample mechanically and chemically complex TMEs to develop next-generation immune-focused therapies to improve patient outcomes.

## Methods

**Human cell culture.** hCD4+ T cells were produced by CD4+ cell isolation and purification from commercially available de-identified whole human blood, supplied by STEMCELL Technologies, Inc. (it is worth noting that details for all cells, mouse models, and reagents are provided in Supplementary Table 1), where blood was tested and negative for Hepatitis C and HIV. CD4 T-cell isolation and purification was performed with EasySep Human CD4+ T Cell Isolation Kit (STEMCELL Technologies, Inc., USA) and then T cells were cultured, activated, and expanded in ImmunoCult-XF T Cell Expansion Medium (STEMCELL Technologies, Inc., USA) with the addition of ImmunoCult Human CD3/CD28/CD2 T Cell Activator and Human Recombinant Interleukin 2 (IL-2, STEMCELL Technologies, Inc., USA) as per the STEMCELL Technologies, Inc. commercial protocol. For CRISPR protocols, hCD4+ cells derived from human whole peripheral blood, were activated and expanded using ImmunoCult Human CD3/CD28/CD2 T Cell Activator (STEMCELL Technologies, Inc., USA), following the manufacturer's recommendation, in ImmunoCult-XF T Cell Expansion Medium (STEMCELL Technologies, Inc., USA) supplemented with Human Recombinant IL-2 (STEMCELL Technologies, Inc., USA). For CRISPR experiments, peripheral blood mononuclear cells (PBMCs) from de-identified healthy human donors were obtained using Trima Accel leukoreduction system (Memorial Blood Centers,

Minneapolis, MN) and further purified using ammonium chloride-based red blood cell lysis and a Ficoll-Paque gradient. All human de-identified human blood samples complied with University of MInnesota Institutional Review Board protocols and the authors complied with all relevant ethical standards for human sample research. CD4+ T cells were isolated from the PBMC population by immunomagnetic positive selection using the MojoSort Human CD4 Nanobeads (BioLegend, San Diego, CA) and an EasySep Magnet (STEMCELL Technologies, Inc., USA). Although fresh CD4+ T cells were readily available from human donors to serve as a platform for T-cell migration, CD8+ T cells from human patients with pancreatic cancer were not available. As such, we confirmed our results with CD8+ cytotoxic T cells from mice with autochthonous PDA (described in the next "Methods" subsection).

**Mouse cell culture.** Mouse CD8+ T (mCD8+) were isolated from tumor-bearing KPC or KPCT mice ($Kras^{G12D/+}$;$p53^{R172H/+}$;$Pdx1$-Cre or $Kras^{G12D/+}$;$p53^{R172H/+}$; $Pdx1$-Cre;$ROSA^{Tdtomato/+}$, respectively, on a mixed background as described[8,15,62] and as approved by the Institutional Animal Care and Use Committee of the University of Minnesota, where the authors complied with all relevant ethical regulations for animal research) using EasySep Mouse CD8+ T Cell Isolation Kit (STEMCELL Technologies, Inc., USA) following the manufacturer's recommendations. All mice were housed in specific pathogen-free conditions and co-housed together as described in the "Mouse tumor slice culture" section. After isolation, mCD8+ T-cell activation and expansion was performed using Dynabeads Mouse T-Activator CD3/CD28 (Thermo Fisher Scientific), following the manufacturer's recommendations in ImmunoCult-XF T Cell Expansion Medium supplemented with Human Recombinant IL-2 for at least 4 days. Magnetic separation was done to separate beads from cells. Cells can be used immediately or can be frozen down in freezing medium (15% dimethyl sulfoxide (DMSO) in fetal bovine serum (FBS)) for future use. It is noteworthy that frozen T cells were given an incubation/recovery period ($t \geq 24$ h) in ImmunoCult expansion medium, supplied with human IL-2 after cell thawing, to avoid the effects of cold exposure shock. All cell work was approved by the University of Minnesota Institutional Biosafety Committee and followed institutional and NIH guidelines.

**High-precision nano-stamps.** A more detailed, step-by-step protocol for this procedure is described elsewhere[69]. Fabrication of elastic ICAM1 or FN nano-patterns is a challenging task due to the van der waals and capillary interactions between the nano-stamp and the printed surface that provoke a collapse of the soft polydimethylsiloxane (PDMS) nano-stamps onto the printed intermediate (glass) surface. To address these issues and achieve high precision of nanopatterns on elastic platforms, we substituted regular PDMS (rPDMS) nano-stamps with composite stamps, veneered with a submillimeter-thick hard PDMS (hPDMS) for non-collapsing, high-definition printing surfaces[11,70]. For the hPDMS preparation protocol, please see "hPDMS formulation" section. To cast the nano-printing surface, we used commercially manufactured polyurethane nanosurfaces as the casting matrices (NanoSurface Biomedical, Seattle, WA). Clean textured nano-surface (NanoSurface Biomedical, Seattle, WA) disks were glued onto the glass platform with SuperGlue (Loctite, USA), silanized with silanizing solution-I as per the commercial protocol (Sigma-Aldrich), coated with ≤0.5 mm hPDMS by gentle spreading with soft Parafilm-made spatula (Hach, USA), cured at 70 °C for 30 min, and subsequently cast with rPDMS to the layer final thickness of 8 mm (1 : 5 curing agent/base ratio, Sylgard-184, Dow Corning Cat# 4019862 and CAS#68988-89-6; see Supplementary Table 1 for details for all reagents). Cured (at 70 °C for ~1 h) composite nano-stamps were peeled and cut into 5 × 5 mm or 1 × 1 cm pieces, and were used as the ready-to-use nano-stamps. To fabricate ICAM1 elastic nano-textures, the Fab anti-Fc antibody fragment protein (Jackson Immunoresearch, USA) was prelabeled with a fluorescent tag and a biotin group, to ensure both its fluorescent visibility in nanopatterns and cross-linking to the streptavidin-functionalized PAA gels, respectively. Briefly, 20 μL of 1 mg/mL antibody sample was incubated for 1 h with 5 μL of (+)-biotin N-hydroxysuccinimide ester (Sigma-Aldrich) and 5 μL of fluorescent tag kit (Alexa Fluor succinimidyl esters, Invitrogen, Molecular Probes) as per the commercial protocols. Labeled protein was then dialyzed overnight in Slide-A-Lyzer MINI Dialysis Device, 7 K MWCO (Thermo Fisher Cat# 69560) overnight at 4 °C in cold phosphate-buffered saline (PBS), then stored at 4 °C in the darkness. Ten-microliter droplets of 0.1 mg/mL labeled antibody solution were then placed atop of the 5 × 5 mm or 1 × 1 cm square micro- or nano-stamps. To ensure a proper coverage and effective stamp surface coating with labeled Fab anti-Fc antibody fragment protein, the protein solution droplet was "sandwiched" between the stamp's printing surface and 15 mm-round glass coverslip (Carolina, USA), which had been baked in the furnace for 5–10 h at 450 °C.

**High-precision 2D nanocontact printing.** A detailed step-by-step protocol for this procedure is described elsewhere[69]. Briefly, using the nano-stamps, we first printed Fab anti-Fc antibody fragment protein nanopatterns onto the "intermediate" glass surface[71], which were then cross-linked to polymerizing PAA gels premixes by their biotin tags to streptavidin-conjugated polyacrylamide (Streptavidin-acrylamide, Thermo Fisher). For that, 7–10 μL of PAA premix of desired G' was polymerized in the "sandwich" manner between an "intermediate" patterned surface

and glass-bottom 35 mm Petri dishes (MatTek Corp., Ashland, MA), activated with 3-(trimethoxysilyl)propyl methacrylate (Sigma-Aldrich) in ethyl alcohol (Pharmco-Aaper) and acetic acid (Fisher Chemical) as per the commercial protocol. For specific *G*'-value formulations, please see the "PAA elastic gels premixes" section. 3-(trimethoxysilyl)propyl methacrylate-functionalized glass surface establishes covalent bonds with the PAA gel upon its curing. Polymerized PAA "sandwiches" were then subjected to hypotonic reversible swelling in deionized water (overnight) for a gentle coverglass release from the PAA gel. The resultant fluorescent PAA nanopatterns of Fab anti-Fc antibody fragment protein were incubated overnight with 1 mg/mL ICAM1-Fc chimeric protein (Sino Biological, China) in cold PBS (Gibco, USA) at 4 °C, rinsed, and used for experiments.

**Fabrication of elastic "2.5D" nanotextures**. A detailed step-by-step protocol for this procedure is described elsewhere[72]. Briefly, nanotextures were cast from PAA gel premixes of chosen shear modulus (*G*'). For specific *G*'-value formulations, please see the "PAA elastic gels premixes" section. As the nanotexture casting master, we used polyurethane-based texturized nanosurfaces (NanoSurface Biomedical, Seattle, WA), which was used to manufacture hPDMS-based molds (please see the hPDMS making protocol in "hPDMS formulation" section). Five to 7 µL droplet of freshly prepared liquid hPDMS premix was sandwiched between polyurethane-based texturized nanosurfaces and a clean activated (baked at 450 °C for 12 h, then ozone or regular air plasma-treated for 5 min) coverglass slide. hPDMS droplet is allowed to spread into a thin layer (5–10 min at ~20 °C), then baked at 70 °C for 1 h. Cured hPDMS "sandwich" is then separated from the molding surface by manual peeling of the hPDMS + glass part of the "sandwich" from the casting polyurethane surface. Peeling process releases the hPDMS surface from the polyurethane casting master, leaving the hPDMS layer attached to the activated glass. The resultant molding nanosurface is then cut in 1 × 1 cm squares by diamond pencil scribbling (on the reverse side of nanosurface) and precoated with biotinylated and fluorescent tag-labeled anti-Fc Fab antibody fragment (Jackson Immunoresearch; 0.1 mg/mL PBS solution, 4 °C, in wet chamber overnight). Alternatively, for fabrication of FN-functionalized elastic nanotextures, 0.1 mg/mL FN protein PBS solutions (bovine plasma FN, Thermo Fisher, Life Science, USA) was used, labeled following the same protocol as for the Fab anti-Fc (see section "High-precision nano-stamps"). After the incubation with the protein solution, the molding nanosurface was gently rinsed in deionized water and blow dried under a filtered air, argon or nitrogen jet. Streptavidin-conjugated polyacrylamide premix of volumes not >0.5 mL was degassed in a vacuum chamber or in an ultrasonication water bath for 1 h. To prevent tetramethylethylenediamine (TEMED) evaporation during the procedure, TEMED is added after the degassing session. Seven to 10 µL droplet of PAA premix of desired *G*'-value was polymerized in the "sandwich" manner between protein-coated nanosurface and glass-bottom 35 mm Petri dishes (MatTek Corp., Ashland, MA), activated with 3-(trimethoxysilyl)propyl methacrylate (Sigma-Aldrich) in ethyl alcohol (Pharmco-Aaper) and acetic acid (Fisher Chemical) in a vacuum chamber. After PAA curing, the resultant textured patterned elastic chip was placed overnight into cold (i.e., at 4 °C) deionized water for PAA-reversible hypotonic "swelling." Then the casting surface was gently peeled from the polymerized PAA surface. For a better release of the sterically interactive nano-mold, hypotonically treated PAA "sandwiches" were optionally ultrasonicated in the water bath for 10 s. Released FN elastic nanosurface is ready for use immediately. Prepared elastic Fab anti-Fc-functionalized PAA nanotextures were incubated with 20 µg/mL human ICAM1-Fc chimeric protein (Sino Biological, China) in cold PBS (4 °C, 12 h), rinsed in PBS three times, and used for the T-cell adhesion and contact-guidance assays.

**hPDMS formulation**. For hPDMS, we mixed 3.4 g of VDT-731 (Gelest, Inc.), 18 µL of Pt catalyst (Platinum(0)-2,4,6,8-tetramethyl-2,4,6,8-tetravinylcyclotetrasiloxane complex solution) (Sigma-Aldrich), and one drop of cross-linking modulator (2,4,6,8-Tetramethyl-2,4,6,8-tetravinylcyclotetrasiloxane) (Sigma-Aldrich). Next, immediately before use, we added 1 g of HMS-301 (Gelest, Inc.) and thoroughly mixed it for 30 s on a vortex mixer[70,73].

**PAA elastic gel premixes**. We chose to control PAA mechanical rigidity via modulation of concentration for both 40% acrylamide (40% AA) base (BioRad) and its cross-linking molecular chain, 2% bis-AA (BioRad) as described elsewhere[69,74]. In addition, streptavidin-acrylamide (Thermo Fisher) was added to the final concentration of 0.133 mg/mL, to enable PAA gels cross-linking with biotinylated proteins of interest. Briefly, for preparation of 50 µL of *G*' = 2.3 and 50 kPa PAA gel premixes, respectively, the following components were mixed: 40% AA: 9.33 and 15 µL; 2% bis-AA: 1.88 and 14.40 µL; 2 mg/mL streptavidin-AA: 3.33 and 3.33 µL; 10× PBS: 5 and 5 µL; deionized milli-Q water: 30 and 11.17 µL; TEMED: 0.1 and 0.1 µL; and 10% ammonium persulfate (APS): 1 and 1 µL. The premix solutions were degassed and stored at 4 °C before use.

**T-cell contact guidance and migration analysis on "2.5D" platforms**. Cell tracking on the 2D and "2.5D" surfaces was performed with the TrackMate plugin in ImageJ. T cells were imaged with differential interference contrast microscopy at 10 s intervals using a ×20 air objective (Nikon Instruments, Japan) and tracking was later performed manually, one cell at a time, at each interval. Cells were

maintained at 37 °C in 5% CO$_2$ (Tokai Hit, Japan) for the duration of the live-cell imaging session. Images were captured at resolutions of 512 × 512 or 1024 × 1024 pixels. For phenotype perturbations, T cells were treated with either 70 nM Taxol or 10 µM Nocodazole.

**T-cell nanotexture invasiveness quantification**. T-cell "in-groove" invasiveness into the nanotextures is measured as the fraction of the T cells' "in-groove" interface to the overall T-cell nanotexture interface. Metrics (left) and measurement (right) of T-cell "in-groove" relative invasiveness are measured as a ratio between area of invasive regions and entire T-cell area at the topography interface:

$$\frac{S_{entire\ cell\ IF}}{S_{invasive\ IF}} \times 100 \qquad (1)$$

**Immunofluorescent labeling for confocal and STED super-resolution microscopy**. T-cell samples were fixed with cold Dulbecco's modified Eagle medium (DMEM) with 4% paraformaldehyde (PFA), followed by 0.1% Triton X-100 in 1% bovine serum albumin (BSA) PBS. F-actin was stained with fluorescent phalloidin (Alexa Fluor phalloidin conjugates, Thermo Fisher Scientific; 10 U/mL) or SiR-Actin (Cytoskeleton, Inc.; 1 µM), after PFA fixation for 1 h in 1% BSA PBS. Chromatin was labeled with 1 : 1000 Hoechst solution (Tocris, USA) and MTs were stained with either Alexa Fluor-conjugated rat anti-tubulin monoclonal antibody (mAb) or an unlabeled version of the same mAb clone YL1/2 (AbCam; 5 µg/mL in 1% BSA PBS, 1 h incubation). All Alexa Fluor fluorescent secondary antibody (Thermo Fisher) labelings were performed at their final concentration of 5 µg/mL for 1 h in 1% BSA PBS.

**Super-resolution imaging and quantification of human T-cell invasiveness on nanoarchitectures**. Super-resolution stimulated emission depletion (STED) microscopy (e.g., Fig. 1A, Supplementary Fig. 1, and Fig. 4A) was performed using a commercial Leica SP8 STED 3× system (Leica Microsystems, Mannheim, Germany), equipped with a white light laser with continuous spectral output between the wavelengths of 470–670 nm and a 592, 660, and pulsed 775 nm STED depletion lasers, to obtain time-gated STED images on 3 hybrid detectors. Given the complexity and varying depth of the sample, we have used the STED WHITE Glycerin objective lens (HC PL APO ×93/1.30 GLYC motCORR) (Leica Microsystems), which is advantageous for depth imaging due to the motorized correction collar, allowing precise and swift adjustment of optical lenses to specimen inhomogeneity. SiR-actin-labeled samples (per commercial protocol) placed in 35 mm culture dishes with bottom coverglass of standard thickness no larger than 1.5 (MatTek Corporation, Ashland, MA) containing 250 µl glycerol (90%) in PBS were imaged sequentially as follows: first sequence STED for SiR-actin (via 647 nm excitation and 660–730 nm emission range) on gated (0.7–6.5 ns time gating) hybrid detector using 775 nm (25% power) as STED depletion laser for best lateral resolution; second and third sequences were confocal settings for Hoechst and Alexa488 (labeling nucleus and surface of the gel), respectively, via two sequential excitations (405 and 488 nm) and two emission ranges (410–465 nm and 495–555 nm), respectively, on gated (0.3–6.5 ns) hybrid detectors. Imaging was performed with a scan speed of 600 lines per second, scanning bidirectionally, a pixel size of 30–35 nm (1024 × 1024 pixels), and 6-line averages, pinhole of 0.7 Airy units, and Z-stacks were collected at 0.140 µm-depth intervals throughout the depth of the sample. We deconvoluted images using Huygens Professional software version 18.10.0 (SVI, Hilversum, NL) with the classical maximum likelihood estimation algorithm. We then inspected and reconstructed 3D data using Clear Volume plugin (FIJI). Still frames were saved and montaged using Adobe Photoshop CC.

Data collection for quantification of cell invasiveness (shown in Fig. 1e) was performed using the instant structured illumination microscopy (iSIM) by an Olympus IX-81 microscope (Olympus, Corp., Tokyo, Japan) equipped with an Olympus UPLAPO-HR ×100/1.5 NA objective, two Flash-4 scientific CMOS cameras (Hamamatsu, Corp., Tokyo, Japan), an iSIM scan head (VisiTech Intl, Sunderland, UK), and a Nano-Drive piezo Z stage (Mad City Labs, Madison, WI). The iSIM scan head included the VT-Ingwaz optical de-striping unit (VisiTech Intl, Sunderland, UK). Image acquisition and system control was through MetaMorph Premiere software (Molecular Devices, LLC, San Jose, CA). Images were deconvolved using the specific for iSIM commercial plugin from Microvolution (Cupertino, CA) in FIJI.

**Atomic force microscopy**. hCD4+ T cells were plated on a glass-bottom dish (Willco Wells) precoated with either ICAM1 (Life Technologies) or 0.01% poly-l-lysine (Sigma-Aldrich) and immersed in culture media solution (Life Technologies). Force spectroscopy AFM experiments were performed using a Bruker Bioscope Catalyst AFM system (Bruker) mounted on an inverted Axiovert 200 M microscope (Zeiss) equipped with a confocal laser-scanning microscope 510 Meta (Zeiss) and a ×40 objective lens (0.95 NA, Plan-Apochromat, Zeiss). The hybrid microscope instrument was placed on an acoustic isolation table (Kinetic Systems). During AFM experiments, T cells were maintained at a physiologically relevant temperature 37 °C using a heated stage (Bruker). A soft silicon nitride tipless AFM probe (HQ:CSC38/tipless/Cr-Au, MikroMasch) was used for T cell's compression. The AFM microcantilevers were pre-calibrated using the standard thermal noise

fluctuations method. The estimated spring constants for microcantilevers used were 0.07–0.1 N/m. After calibration, the AFM probe was moved on top of a rounded T cell. Five to ten successive force curves were performed on each T cell. The deflection set-point was set to 20 nm yielding applied forces between 1.5 and 2 nN.

**Analysis of AFM data: determination of T cell's cellular surface tension**. All AFM force–distance curve measurements were analyzed to calculate the cellular surface tension. For curves fitting, indentation depths between 0 and 400 nm were relatively consistent in yielding good fits ($R^2 > 0.9$). Curves with poor fits $R^2 < 0.9$ were discarded from the analysis. In addition, we discarded noisy force curves and/ or curves that presented jumps possibly due to cantilever and plasma membrane adhesion, slippage, or very weakly adhered and moving cells. T-cell surface tension ($T$; pN/μm) was computed by fitting each recorded force–distance curve with the surface tension formulae described in refs. [48,53], which defines the force balance relating the applied cantilever force with the pressure excess inside the rounded cells and the corresponding surface tension:

$$T = \frac{k_c}{\pi}\left(\frac{1}{(z/d)-1}\right) \tag{2}$$

where $T$ is the cellular surface tension, $k_c$ is the AFM cantilever spring constant, $Z$ is the $Z$-piezo extension, and $d$ is the cantilever mean deflection. In addition, the T cells intracellular hydrostatic pressure ($P$; Pa) was calculated by using Laplace's law for spheres:

$$P = \frac{2T}{R}, \tag{3}$$

where $P$ is the intracellular hydrostatic pressure and $R$ is the initial radius of the T cell.

**T-cell genome engineering with CRISPR**. Single-guide RNAs (sgRNAs) were designed to human *ARHGEF2 (GEF-H1)* using the Synthego CRISPR design tool (https://design.synthego.com/). The top two recommended guides were obtained as modified sgRNAs (Synthego, Menlo Park, CA). Both sgRNAs were screened in K562 cells and the most efficient gRNA (Sequence: 5′-GAGGUGCCCAUUG-GUAUAGC-3′) based on KO score was used in subsequent experiments with primary human T cells. For T-cell culture, T cells were maintained in OpTmizer CTS T cell Expansion serum-free media (SFM) containing 2.5% CTS Immune Cell Serum Replacement (Thermo Fisher, Waltham, MA), L-glutamine, penicillin–streptomycin, *N*-acetyl-L-cysteine (10 mM), rhIL-2 (300 IU/mL), rhIL-7 (5 ng/mL), and and rhIL-15 (5 ng/mL), at 37 °C with 5% $CO_2$. T cells were activated with Dynabeads Human T-Activator CD3/CD28 (Thermo Fisher, Waltham, MA) at a 2 : 1 bead : cell ratio for 48 h prior to electroporation. T cells were maintained at ~$1 \times 10^6$/mL in normal tissue culture flasks for experiments optimizing editing efficiency. For T-cell electroporation, after 48 h, Dynabeads were magnetically removed and cells were washed with PBS once prior to resuspension in the appropriate electroporation buffer. Primary human T cells ($1 \times 10^6$) were electroporated using the 4D-nucleofector (Lonza, Basel, Switzerland) and a P3 Primary Cell 4D-Nucleofector X Kit (V4XP-3032). CleanCap Cas9 mRNA (1.5 μg; TriLink Biotechnologies, San Diego, CA) and 1 μg of modified gRNA were added to $1 \times 10^6$ cells in 20 μL of the recommended electroporation buffer. The mixture was electroporated using the program EO-115. Cas9 mRNA alone was used as a control for all conditions. Following electroporation, T cells were allowed to recover in antibiotic-free medium at 37 °C, 5% $CO_2$ for 20 min, then cultured in complete CTS OpTmizer T cell Expansion SFM as described above. Genomic DNA was taken from T cells 7 days post electroporation by spin column-based purification. Cas9 efficiency was analyzed on the genomic level by PCR amplification of CRISPR-targeted loci (forward sequence: 5′-AGGGAGATGAGTGGCAACAG-3′; reverse sequence: 5′-CAGCTGGGGATCAGAGAGAA-3′), Sanger sequencing of the PCR amplicons (Eurofins Genomics, Louisville, KY), and subsequent analysis of the Sanger sequencing traces using the ICE web app developed by Synthego (https://ice.synthego.com/).

**Rho activation assay**. Levels of active Rho were evaluated in human T cells using the G-LISA assay (Cytoskeleton, Inc., Cat # BK124). T cells were treated with vehicle (DMSO) or 10 μM Nocodazole for 20 min. The assay was performed according to the manufacturer's instructions.

**Preparation of 3D collagen matrices**. Collagen matrices preparation was adapted from ref. [75] with modifications. Briefly, high concentration rat-tail collagen-I (VWR) was neutralized with a 1 : 1 ratio of 100 mM HEPES (Thermo Fisher Scientific) in 2× PBS and completed to a final concentration of 3 mg/mL with DMEM (Corning) supplemented with 10% FBS (which introduces FN). The mixture was allowed to sit in ice for 5 min, after which 350 μL was pipetted into a standard 24-well plate. The gels were left polymerizing for 20 min, in the hood, at room temperature and then incubated at 37 °C with 5% $CO_2$ for 3 h and subsequently overlaid with DMEM (Corning), and incubated overnight at 37 °C with 5% $CO_2$. For some instances of linear microscopy analysis, we imaged the collagen matrix

via fluorescence instead of second harmonic generation (SHG) imaging. To fluorescently label collagen-I matrix, we added 1 : 20–1 : 100 (volume) of 1 mg/mL fluorescent tag (Alexa Fluor succinimidyl esters of the desired excitation/emission spectrum, Invitrogen, Molecular Probes) to the gel premix immediately before polymerization.

**T-cell migration in 3D collagen matrices**. Viable hCD4+ or mCD8+ T cells ($3 \times 10^5$) were stained with 1 μM of CellTracker Green CMFDA (5-chloromethyl-fluorescein diacetate; Thermo Fisher) for 5 min, protected from light, at 37 °C. Lymphocytes were centrifuged at $300 \times g$ for 5 min and washed twice in L-15 plus 1% FBS and resuspended in the same media to a volume of 200 μL. Excess overlaid media from the collagen matrices was removed completely prior to seeding T cells. The $3 \times 10^5$ T cells in 200 μL were gently pipetted on top of the collagen matrix, gently swirled to evenly distribute the cell suspension, and incubated for 30 min at 37 °C and 0% $CO_2$ to allow lymphocytes to infiltrate the gel matrix. After incubation, the remaining media on top of the matrix was removed and washed once, to remove cells that did not infiltrate/adhere to the gel. Gel was carefully detaching from all sides using a pipette tip and transferred to a 35 mm dish using tweezers. A slice anchor (Warner Instruments) was used to hold the gel in place and was overlaid with 5 mL of L-15 media plus 1% FBS in control (DMSO), 70 nM Taxol, 10 μM Nocodazole, 5 μg/mL RhoA Activator II (Cytoskeleton, Inc.), or 2 μg/mL RhoA Inhibitor I (Cytoskeleton, Inc.) conditions and imaged immediately after designated incubation time as per commercial protocol (Cytoskeleton, Inc.). Briefly, for RhoA activation or inhibition, prior to staining T cells with CMFDA, T cells were resuspended in serum-free Roswell Park Memorial Institute (RPMI) with 5 and 2 μg/mL of activator and inhibitor for 3 and 4 h, respectively.

**Mouse tumor slice culture**. Genetically engineered $Kras^{LSL-G12D/+};p53^{LSL-R172H/+};Pdx1-Cre$ (KPC) and fluorescent reporter KPC mice [$Kras^{LSL-G12D/+};p53^{LSL-R172H/+};Pdx1-Cre;ROSA^{Tdtomato/+}$ (KPCT)] were used as a faithful mouse model for pancreatic cancer[4,15,62] as approved by the Institutional Animal Care and Use Committee of the University of Minnesota, where the authors complied with all relevant ethical regulations for animal research. Mice are on a mixed background and were housed in specific pathogen-free conditions with a 12 h light/12 h dark cycle, and co-housed together. Freshly explanted KPC and KPCT tumors were collected at the endpoint and placed in ice-cold sterile PBS with soybean trypsin inhibitor (STI; ATCC). Agarose gel (1.5%) (Genemate) was prepared and super glued (Loctite, USA) onto the vibratome (Vibratome Company) cutting stage to create a support for the tumor, which was super glued directly in front of the agarose gel. The stage was overlaid with ice-cold PBS. The sectioning parameters were as follows: speed 3 mm/s, amplitude 8 mm, and 350 μm thickness. Slices were placed in ice-cold sterile PBS with STI for transport and subsequently cultured (4 days maximum), with modifications from a published protocol[76]. Here, multiple slices were transferred, using tweezers, and placed flat on 0.4 μm, 30 mm-diameter cell culture inserts (Millipore Sigma) precoated with collagen gel mixture, as described above, but completed with 1× PBS instead of growth media, in a six-well plate, with RPMI 1640 supplemented with 10% FBS, 1% penicillin and streptomycin, 14.5 mM Hepes, 5 μg/mL plasmocin (Invivogen), and 10 μg/mL of STI in a 37 °C humidified incubator at 5% $CO_2$. Single slices can be cultured similarly in a 0.4 μm, 12 mm-diameter cell culture inserts (Millipore Sigma) in a 24-well plate. The culture media was changed daily.

**T-cell migration in tumor slices**. KPC slices were transferred to a 24-well plate and cultured in L-15 media supplemented with 10% FBS, 1% penicillin and streptomycin, 5 μg/mL plasmocin, and 10 μg/mL of STI with 5 μM of CellTracker Red CMTPX (Invitrogen) for 15–20 min at 37 °C with 0% $CO_2$ incubator. KPCT slices were not stained with CellTracker Red, as carcinoma cells can be visualized via red fluorescence. Slices were washed twice in the cell tracker-free media. Individual slices were then transferred using tweezers to the 12 mm culture inserts in the 24-well plate. mCD8+ T cells (1–2 × 10⁵) were stained, as described above, and concentrated in 100 μL of L-15 media supplemented with 10% FBS, and immediately added onto the slice, in the inserts, as to concentrate the cells on the slice and let the cells adhere/migrate on the tissue for 1 h at 37 °C with 0% $CO_2$ incubator. Slices were then handled with tweezers and gently washed in the same media to remove excess T cells that did not migrate/adhere into the tissue and transferred to a 3.5 cm dish. A slice anchor was used to hold the tissue in place and immediately overlaid with 5 mL of L-15 media plus 1% FBS in control (DMSO), 70 nM Taxol, and 10 μM Nocodazole conditions, and imaged immediately.

**Multi-photon microscopy of 3D live-cell imaging and analysis of 3D cell migration**. Cell migration data were collected by imaging using a custom-built multi-photon laser-scanning microscope (Prairie Technologies/Bruker) using a Mai Tai Ti:Sapphire laser (Spectra-Physics) to simultaneously generate MPE and SHG to visualize cells and collagen, respectively, at an excitation wavelength of 880 nm with a custom-built temperature-controlled stage insert, as described[8,75]. Briefly, time-lapse imaging of T cells inside a 3D collagen matrix and on tumor slices was obtained by creating two-channel (T cells and collagen) or three-channel (T cells, collagen, and KPC-tissue/KPCT-carcinoma cells) Z-stacks of 75 μm depth at 5 μm steps, at each stage position, every 1.5 min over at least 45 min of imaging.

Five dimensions (x, y, z, t,, in two/three channels) time-lapse images were post-processed in Fiji[77], drift corrected using 3D correction plugin[78], and cell tracking was performed using TrackMate Plugin[79] in Fiji. Quantification and analysis of tracks was performed as previously described in refs. [8,75]. Here, cell trajectories were fit to a PRWM[80,81] using the method of overlapping intervals[82] using MATLAB (Mathworks) as previously described[75]. The mean squared displacement for a cell over a time interval $t_i$ was obtained by averaging all the squared displacements $x_{ik}$ such that

$$\bar{x}_i = \frac{1}{n_i} \sum_{k=1}^{n_0} \overline{x_{ik}} \tag{4}$$

$$n_i = N - i + 1 \tag{5}$$

where $n_i$ is is the number of overlapping intervals of duration $t_i$ and $N$ is the total number of intervals. Mathematically, the PRWM can be written as:

$$MSD(t) = 2S^2P[t - P(1 - e^{\frac{-t}{P}})], \tag{6}$$

where $S$ is the migration speed and $P$ is the persistence time. The motility coefficient is given as:

$$\mu = \frac{S^2 P}{n_d} \tag{7}$$

where $n_d$ is dimensionality. As the model was fit separately into the three orthogonal directions, we obtained motility, speed, and persistence times for $x$, $y$, and $z$ directions; therefore, $n_d = 1$ in each case.

For total speed of each cell, we took the square root of the squared sum of each speed in the $x$, $y$, and $z$ directions as follows:

$$S = \sqrt{S_x^2 + S_y^2 + S_z^2} \tag{8}$$

For total motility of each cell, we added the different motilities from each direction as follows:

$$\mu = \mu_x + \mu_y + \mu_z \tag{9}$$

**Statistical analysis**. Pairwise comparisons were analyzed using a one-sided $t$-test. Multiple groups were compared by one-way analysis of variance, followed by the Tukey's post hoc analysis. Figure legends indicate which statistical test was performed for the data. Statistical analysis was performed using either KaleidaGraph 4.5.4 (Synergy Software) or Prism 7b (GraphPad Software, Inc.). The exact $p$-values are indicated on the plots, unless the $p < 0.0001$, where 0.0001 is a cut-off lower limit for Kaleidagraph and Prism softwares. Sample size $n$ for each comparison is reported in the corresponding plots (i.e., $n$ reflects the number of measured individual cells or the cells' movement steps per given time interval). Number of replicates is three or more, unless specified otherwise. Data are shown as box and whiskers diagrams: 95% confidence interval, first quartile, median, third quartile, and 95% confidence interval.

**Reporting summary**. Further information on research design is available in the Nature Research Reporting Summary linked to this article.

## Data availability

The authors declare that the data supporting the findings of this study are available within the paper, the Supplementary Information, and in the Source Data file, and any data can be made further available upon reasonable request. Source data are provided with this paper.

## Code availability

Custom code is included in the Supplementary Materials file. Source data are provided with this paper.

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

## Acknowledgements

P.P.P. and this work was supported by the NIH (R01CA181385 to P.P.P., U54CA210190 University of Minnesota Physical Sciences in Oncology Center to P.P.P., a supplement to R01CA181385 to E.D.T., and NIAID training grant T32AI997313 to E.J.P.) and by a Research Scholar Grant RSG-14-171-01-CSM from the American Cancer Society. This work was also supported by the Randy Shaver Research and Community Fund (P.P.P.), grants from the UMN Institute for Engineering in Medicine (P.P.P.), and the Children's Cancer Research Fund (B.S.M and B.R.W.). A.C.R. was supported by the NIH Distinguished Scholars Program and the NIH Intramural Research Program of the National Institute of Biomedical Imaging and Bioengineering. A.S.Z. was supported by the NIH Intramural Research Program of the National Heart, Lung, and Blood Institute. We thank the University of Minnesota Imaging Center (UIC) and UIC staff, particularly Dr. Guillermo Marqués, for helpful assistance (https://med.umn.edu/uic). The content of this work is solely the responsibility of the authors and does not necessarily represent the official views of the NIH or other funding agencies.

## Author contributions

E.D.T. and P.P.P. participated in the scientific conceptualization and design of the study. E.D.T. engineered elastic "2.5D" nanoplatforms and developed the T-cell biophysical phenotypes balance model and T-cell migration contact-guidance assays. N.J.R. developed T-cell 3D migration assays and live tumor imaging. P.P.P. developed the concept of engineering T cells to be mechanically optimized. A.C.R. conducted T-cell atomic force microscopy tests. E.D.T., N.J.R., A.S.Z., and P.P.P. designed experiments. E.D.T., N.J.R., A.C.R., M.K.C., E.J.P., K.Y., W.S.L., A.S.Z., B.R.W., and P.P.P. conducted experiments

and data analysis. E.D.T., N.J.R., A.C.R., V.P., E.A.E., E.J.P., K.Y., W.S.R., B.R.W., B.S.M., A.S.Z., and P.P.P. analyzed data. E.D.T., M.K.C., E.J.P., N.J.R., K.Y., W.S.L., and B.R.W. generated unique reagents and their validation. E.D.T., A.S.Z., A.C.R., and N.R.J. performed quantitative analysis and algorithm implementation. P.P.P. and B.S.M. secured funding. E.D.T., N.J.R., and P.P.P. wrote the manuscript. All authors read and contributed comments to the final manuscript. P.P.P. oversaw all aspects of the study.

## Competing interests

The authors declare no competing interests.
