## [Peer Review File · Nature Communications]

REVIEWER COMMENTS

Reviewer #1 (Remarks to the Author):

The manuscript by Tabdanov et al. addresses a critical hurdle in cancer immunotherapy: the ability of T cells to traffic into solid tumor microenvironments. The manuscript uses novel methodologies and is clearly written and suggests potentially impactful approaches for cancer therapeutics and adoptive T cell transfer. The authors use multiple in vitro systems for examination of T cell migration and extra-cellular matrix interactions. Multiple drugs are used, including those relevant to human cancer. One question that the manuscript is why the authors chose human CD4+ and murine CD8+ (and not the same subset from each species or both subsets from both species), which should be mentioned somewhere in the text. Lastly, there is a typo on pg. 8: "were" should be "where".

Reviewer #2 (Remarks to the Author):

In their manuscript titled "Engineering T cells to enhance 3D migration through structurally and mechanically complex tumor microenvironments", Tabdanov and colleagues seek to better understand the molecular mechanisms driving T cell movement through extracellular matrix environments like those found in and around pancreatic tumors. They use a range of models to investigate this research question, specifically examining the role of microtubules in cell migration through substrates with varying stiffness and dimensionality. Utilizing their innovative "2.5 D" nanotextured substrates, the authors conclude their cells form two distinct types of protrusions: lamellipodia that form on ridges and pseudopods that form in trenches or grooves in the pattern. By varying substrate stiffness and treating cells with either nocodazole or taxol, the authors report that microtubules play a regulatory role in cell migration in or on stiff substrates. Importantly, the same trend of increased migration was seen in nocodazole-treated T cells within tumor slices, as well as 3D collagen. Finally, the authors speculate that this increase in speed is due to the release of GEF-H1 upon microtubules depolymerization. The authors conclude that release of GEF-H1 leads to increased Rho activity and intracellular pressure that switches cells to a bleb/pseudopod-based migration through 3D collagen.

The migratory plasticity of T cells is relatively understudied compared to other cell types, so the manuscript is timely and should be of general interest to a wide audience. Overall, the data is quite clear and appropriately controlled, and the figures are easy to understand. However, while the paper has good evidence to support some of the authors' conclusions with regard to the potential role of microtubule stability in governing T cell speed, there is no data presented that the cells are actually changing the types of protrusions they are using to migrate. Without direct evidence that the T cells are migrating using distinct protrusions in each assay, a major conclusion of the manuscript relies solely on speculation. This is a critical weakness that needs to be addressed, along with some additional points noted below.

Comments:

There are no data presented that clearly show the T cells are using distinct types of protrusions to migrate across the ridges versus in the grooves of the 2.5D model, within the 3D collagen, and the tumor slices. Without this experimental data, any conclusions that microtubule stability and contractility are regulating the types of protrusions formed and thereby cell speed rely on excessive speculation. To mitigate this concern, the authors need to demonstrate the protrusions formed in each environment/condition are truly distinct from one another. This could be accomplished by showing the presence or absence of specific protein markers (such as Arp2/3 or cortactin in the case of lamellipodia), demonstrating the protrusion morphology and dynamics are distinct, and showing the protrusion dynamics can be distinguished by their sensitivity to select chemical inhibitors (such as the Arp2/3 or myosin II inhibitor). The authors might think of a more effective way to accomplish this goal, but the important thing is they clearly show that the cells are using distinct types of protrusions to support one of their main conclusions.

Some of the main figures include rather idealized schematics that are not supported by the

evidence within the figure, mainly for the reasons discussed above. For example, in Figure 1, panels B and D schematics show that the cells form lamellipodia and pseudopodia/blebs on the ridge in the grooves/3D collagen, respectively. These should be replaced by the primary data used to construct the figures in panel E (currently found in Supplemental Figure 1). It would be more appropriate for all the idealized schematics (Fig. 1B, C, and D; Fig. 2D, and Fig. 3C) to be condensed, moved to the end of the paper, and presented as a speculative models unless there are data included in the manuscript that directly supports them.

Some additional evidence could also reinforce the conclusions leading to their overall model of RhoA activation triggering rapid amoeboid migration. Specifically, can the change in GEF-H1 localization be confirmed following nocodazole addition and can they demonstrate that the abundance of RhoA-GTP increases in the same conditions and that increase requires GEF-H1 expression?

Including a representative video of cell migration or protrusion dynamics for some of their key results seems like a great opportunity to directly show the reader the dramatic changes they are measuring.

Lamellipodia are a type of pseudopodia, thus labelling the in-groove protrusions as pseudopodia is not particularly informative. If their further experiments can clearly show they are a bleb-based protrusion, the more specific label should be applied where appropriate.

How do the protrusions identified in the 2.5D model relate to each other in a 3D context, i.e. what is the physiological relevance of the "2.5 D" substrates? Can the T cells migrating in 3D collagen use both types of protrusions simultaneously?

To apply Laplace's law for spheres to calculate cellular hydrostatic pressure from tension the cell needs to be spherical. Given all of the T cell images in the manuscript show the cells are not spherical, how can the authors justify this approach to calculate hydrostatic pressure?

What evidence is there the nocodazole and taxol treatments are depolymerizing and stabilizing microtubules in these experiments, respectively? Including movies that show what treatments do to protrusions and IF images that show how treatments rearrange microtubules could help to illustrate the effectiveness of the treatments.

The emphasis on engineered cells in the title seems incongruent with the fact they are only used in the final figure. I suggest the authors think about using a title that more reflects the major focus/findings of the paper.

Reviewer #3 (Remarks to the Author):

Tabdanov et al. report on the role of contractility in T-cell phenotype and migration. The work presented in this manuscript evaluated T-cell migration and shows that inducing microtubule instability promotes migration. This work suggests a rationale that engineered T-cells may improve T-cell infiltration in malignancies, which could have significant therapeutic implications. I commend the authors on their well thought-out, hypothesis driven experiments.

The work has nicely investigated the mechanistic and mechanosensing behavior of T-cells using 3D ("2.5D") models. The investigators have utilized both pharmacological and genomic intervention strategies to corroborate their findings and demonstrate that the manipulation of microtubule contractility leads to effective migration of T-cells in a 3D tumor microenvironment, and that this process involves the GEF-H1/RhoA pathway. The experimental design uses 3D nanotextured elastic platforms and live tumors in an attempt to define a mechanical optimum for migration that can be perturbed by manipulating microtubule stability. The authors report that "microtubule instability, leading to increased RhoA pathway-dependent cortical contractility promotes migration, while clinically used microtubule-stabilizing chemotherapies profoundly decrease effective

migration". They hypothesize that T-cell migration can be improved by rational manipulation of microtubule contractility to improve anti-tumor immunity. They then describe two types of T-cell motility, a desirable amoeboid phenotype with low adhesion pseudopodia and a sub-optimal mesenchymal-like phenotype with adhesive spreading. Using RhoA activation and inhibition they demonstrate the role of RhoA signaling in T-cell contractility. Furthermore, knockout of a key downstream molecule of Rho-mediated contractility, GEF-1, reduced contractility. Surprisingly, this was observed upon treatment with a microtubule destabilizer while contractility increased in the absence of the GEF-1 molecule. They reproduced the relationship between T-cell contractility and migration using tumor tissue in place of the artificial matrix. The authors conclude that pharmacologic manipulation of microtubule contractility has the potential to improve immunotherapy. This is a very novel study of how the physical properties of T-cells and T-cell migration are controlled by microtubules. The role of microtubules in T-cell migration is not surprising. However, the elegant studies presented in this manuscript establish new approaches for researching the precise mechanisms involved. In spite of the strong and novel physicochemical analysis, the report is rather shallow in T-cell and cancer biology. While focusing on the intratumoral migration of T-cells, the manuscript does not address the barriers to T-cell infiltration into tumors nor the adhesion molecules and chemokines involved. Among the barriers to T-cell infiltration is the role of tumor vascularity. For example, chordoma and chondrosarcoma are two sarcomas that are both quite avascular. Because of this avascularity, patients who have either of these two sarcomas currently have no chemotherapy options. This study also does not consider the immunotolerant microenvironment of the tumor. The language used to describe the KPC mouse model is confusing and probably incorrect; for example, reference to the KPC mouse being a model of metastatic pancreatic cancer. Finally, the study provides no in-vivo evidence that microtubule destabilization is protective in cancer, even in a simple animal model. The authors also do not address the fact that tumor cells mimic lymphocyte migration characteristics to metastasize; hence their approach may enhance tumor metastasis.

The manuscript is well written, the overall experimental design is sound, the data are clearly presented, and the interpretation of the results is rational. The discussion could be made easier to follow for readers who are not in the field. Additional experiments are needed to define the translational and disease significance of this study.

This study would become much stronger, and would have a greater impact if the following revisions were made. It is the recommendation of this reviewer that the authors make them:

1. The introduction nicely summarizes the current understanding of T-cell migration, as well as areas where further investigation is needed, thus offering rationale for the subsequent work. While the final paragraph of the manuscript offers additional rationale, I would suggest adding a statement or two in this paragraph that highlights the overall hypothesis so that readers are reminded of the motivation for the experiments.
2. The results are well organized, and there is significant discussion within the results section, including introductory information as well as conclusions that the authors have drawn based on their results. The actual data itself becomes lost. I suggest revisions to this section to make their important findings more clear, in addition to simply referencing figures. For example, when discussing T-cell migration directionality on various nanotexture rigidities, the authors state that quantification per every 10 second step demonstrates significant differences in T-cell migration directionality. Providing these data and stats in the text would help distinguish actual results from this experiment from the discussion offered in this section. A similar example could be given for each sub-section within the results section.
3. The experimental design includes only a single dose of each of the agents Nocodazole and Taxol. Additional dose levels would strengthen the data set for this experimental section.
4. Since the results and discussion sections are separate sections, the interpretations and comparisons to published works in the results section should be moved to the discussion section.
5. Key experiments showing the migration of T-cells in the presence of pancreatic cancer cells should be repeated using primary cells, preferably derived from organoid cultures or Patient Derived Xenograft (PDX) models having different immunomodulatory profiles (e.g., expression of different chemo-attractant or immunosuppressive molecules).
6. It will be important to use cells isolated from pancreatic cancer patients for the initial characterization of the nano-platform and define the role of GEF-H1 in the migration of those cells.

7. The rationale of looking at GEF-H1 is not clearly defined. Why do the authors focus on this guanine exchange factor (GEF)? There are other GEFs expressed in T-cells that can activate Rho. This should be clarified.
8. The theoretical potential for enhancing metastatic tumor cell migration needs to be discussed either by published reference articles or additional experiments.
9. The report is written in a very difficult prose and contains much "jargon". It can surely be translated into more easy to read English. In its present format, it would be of interest to a rather limited audience and thus be more suitable for a specialized journal.

RESPONSE TO REVIEWERS

Re: Article number NCOMMS-20-22556-T: "Engineering T cells to enhance 3D migration through structurally and mechanically complex tumor microenvironments"

We are pleased to submit a significantly revised version of our manuscript. We are grateful to the reviewers for their careful consideration of our manuscript and the excellent suggestions to improve and clarify the work. We have endeavored to address all of the concerns raised. We have revised the text and added additional data and analysis. We believe these efforts have extended the rigor and impact of the work. We present below a detailed point-by-point response to the reviewers' questions and comments.

RESPONSE TO REVIEWER 1

We are grateful to the reviewer for their encouraging and insightful comments. We sincerely appreciate the thought and rigor of the comments during the evaluation of our work. A number of important points and questions were raised that we have endeavored to address:

1) The manuscript by Tabdanov et al. addresses a critical hurdle in cancer immunotherapy: the ability of T cells to traffic into solid tumor microenvironments. The manuscript uses novel methodologies and is clearly written and suggests potentially impactful approaches for cancer therapeutics and adoptive T cell transfer. The authors use multiple in vitro systems for examination of T cell migration and extra-cellular matrix interactions. Multiple drugs are used, including those relevant to human cancer. One question that the manuscript is why the authors chose human CD4+ and murine CD8+ (and not the same subset from each species or both subsets from both species), which should be mentioned somewhere in the text. Lastly, there is a typo on pg. 8: "were" should be "where".

Thank you for your support for our work and highlighting two areas to increase clarity and accuracy. We utilized human CD4+ T cells as a general platform and verified our results with CD8+ T cells from tumor bearing mice. We have attempted to obtain CD8+ T cells from human pancreatic cancer patients (and have an approved IRB in place), however, to date, no patients have consented to the additional blood draws limiting our ability to utilize fresh human CD8+ T cells from patients with cancer. In contrast, fresh human T cells from healthy donors can be obtained commercially. We have clarified this in the both Results and Methods sections of the manuscript. We have also corrected the typographical error on page 8.

RESPONSE TO REVIEWER 2

1) There are no data presented that clearly show the T cells are using distinct types of protrusions to migrate across the ridges versus in the grooves of the 2.5D model, within the 3D collagen, and the tumor slices. Without this experimental data, any conclusions that microtubule stability and contractility are regulating the types of protrusions formed and thereby cell speed rely on excessive speculation. To mitigate this concern, the authors need to demonstrate the protrusions formed in each environment/condition are truly distinct from one another. This could be accomplished by showing the presence or absence of specific protein markers (such as Arp2/3 or cortactin in the case of lamellipodia), demonstrating the protrusion morphology and dynamics are distinct, and showing the protrusion dynamics can be distinguished by their sensitivity to select chemical inhibitors (such as the Arp2/3 or myosin II inhibitor). The authors might think of a more effective way to accomplish this goal, but the important thing is they clearly show that the cells are using distinct types of protrusions to support one of their main conclusions.

The Reviewer's comment motivated us to step back and re-examine the structure and presentation of our work to more clearly demonstrate the distinct behaviors. As a consequence of this, we have edited the text and added new data and analysis in both the primary and supplementary figures, and added two key movies. In particular, we note that phenotypes occur simultaneously and the balance between them can be shifted. In Supplementary Figure 1 we show the distinct "in-groove" and "on-ridge" phenotypes and how the balance can become more dominant for "in-groove" interactions based on nanogroove stiffness or altered microtubule stability. In the new **Supplementary Figure 2** we show detailed differences in phenotype resulting from microtubule destabilization and also add data showing that inhibition of Arp2/3 blocks the formation of flat "on-ridge" lamellipodia. Following inhibition of Arp2/3, lamellipodium no longer forms and the cell linearizes to a more pointed spindle-like morphology, oriented along both nanogroove and nanolines. In the new **Supplementary Figure 3** we present time-course data that shows the distinct phenotypes under key conditions, Control vs. Nocodazole treated cells, and also now include two Supplementary Movies of these conditions showing the distinct phenotypes under each condition in migrating cells over time. In **Figure 2** we also added new super-resolution imaging showing the distinct phenotypes.

For instance, we have edited the 2nd paragraph of the Results text to better present our primary data and the additional data introduced during the revision such as:

"Examination of T cell behavior revealed that these platforms are sterically interactive as they concurrently allow for both T cell "in-groove" amoeboid invasiveness (i.e. sterically-interactive pseudopodia) into textures and more mesenchymal-like "on-ridge" pseudopodia (i.e. lamellipodia) spreading behavior (see co-existing "on-ridge" and "in-groove" phenotypes in **Figure 1 and Supplementary Figures 1, 2, and 3**). Indeed, the interactions between activated primary human T cells and elastic nanotextures reveal a partial T cell interface invasiveness into the nanogrooves depth (i.e. "in-groove" plane; **Figure 1a, b and Supplementary Figures 1 and 2**). The remaining "noninvasive" part of the T cell-nanotexture interface is structured more like a flat lamellipodium spread atop of the nanotextured plane (i.e. more mesenchymal-like at the "on-ridge" level; **Figure 1b and Supplementary Figures 1, 2, and 3**). Note, we define the mesenchymal-like phenotype for T cells as lamellipodium-driven flat adhesive spreading atop the nanoridges, similar to that observed during migration on flat environments. We also note that inhibition of Arp2/3, a key regulator of lamellipodia dynamics, confirms the presence of lamellipodia in T cells as inhibition abolishes the lamellipodia, resulting in a spindle-like phenotype (**Supplementary Figure 2**). This is consistent with carcinoma cell behavior...However, since the nanogroove platforms developed here facilitate partial interactions that allows for both highly directed "in-groove"-confined protrusions and less directed "on-ridge" 2D mesenchymal-like lamellipodium protrusions (**Figure 1a,b**), i.e. a superposition of amoeboid and mesenchymal functional phenotypes (**Figure 1b**), they allow us to define the mechanical regulators that drive the amoeboid-mesenchymal plasticity balance (**Figure 1d,e**) and develop design criteria"

And in the 6th paragraph we edited the text to read "Moreover, as expected from our analysis of T cell "in-groove" vs. "on-ridge" interactions with nanotexture (**Figure 1**), in control conditions (+DMSO), T cell migration speed displays a direct and proportional correlation to the degree of T cell "in-groove" invasiveness (i.e. the amoeboid-mesenchymal phenotype balance), with significantly slower migration as substrate stiffness increases (i.e. shifted toward a larger contribution from the "on-groove" mesenchymal phase; **Figure 2a-c, +DMSO**). This again suggests that softer environments result in higher T cell cortical contractility that more robustly facilitates fast directed migration. That is, softer environments result in weaker T-cell-to-nanotexture "on-ridge" adhesion and spreading, allowing for mechanical dominance of T cell cortical contractility that promotes amoeboid dynamics and more robustly facilitates fast directed migration."

We also noted distinct phenotypes following treatment with microtubule targeting agents, in particular a shift to more "in-groove" invasiveness following Nocodazole treatment (**Figures 1 and 2 and Supplementary Figures 1, 2 and 3**). We have significantly extended this data in the revised manuscript (e.g. added data to Figure 2 and added Supplementary Figures 2 and 3 and key Supplementary Movies). We have edited the text to reflect

these changes and better clarify our observed phenotype behavior, e.g. “Indeed, MT disruption results in a shift towards more “in-groove” protrusions (i.e. tilts to the phenotype balance toward a more amoeboid phenotype; **Figure 2d and Supplementary Figures 1-3, and Movies 1-4**) and highly directed migration from contact guidance across all substrate stiffnesses on both ICAM1 and FN (**Figure 2a,b and Supplementary Figure 4a,b**)”.

We have also modified the text associated with our conclusion that blebbing is the initiating event for “in-groove” protrusions. The initial text now reads: “From a mechanobiology perspective, the fact that we did not observe any filopodia in nanogrooves of any stiffness suggests that “in-groove” amoeboid pseudopodia formation is primarily initiated by cytoplasm hydrostatic pressure³¹ generated by high cortical actomyosin contractility. Cytoplasm pressure-driven off-cortex plasma membrane peeling³²⁻³⁴ results in budding of the invasive plasma membrane blebs, i.e. pseudopodial precursors (**Figure 1b**), which then rebuild an actomyosin cortex in the new pseudopodia extensions³²⁻³⁴. Therefore, we hypothesized that in-groove pseudopodia formation dynamics can be inherently antagonized by mechanical tension from the cell-substrate adhesion interface from the “on-ridge” pseudopodia, i.e. the mesenchymal-like T cell adhesion mechanisms (**Figure 1b,d**), similar to observations in carcinoma cells¹¹. This suggests that the mechanical properties of the substrate, and the related intracellular mechanics, play a key role in governing directed motility of T cells. To explore this regulation...”. In particular, we note that we did not observe any filopodia in the soft or stiff nanogrooves. In fact we have not registered filopodia in the nanogrooves in any conditions, except for very infrequent findings following Taxol treatment, where “in-groove” protrusions are low and contact guidance along the nanotextured is minimal (**Figure 2**). Further, filopodia are known to depend on mechanically regulated adhesion that enhances cell-substrate interactions. We, however, observe strongest T cell guidance on the soft nanotextures, which suggests that topographic contact guidance from initiators of “in-groove” protrusions is driven by steric interactions and not mechanical reinforcement in filopodia. As such, the literature suggests that high cortical contractility is driving small blebs that are precursors to the “in-groove” pseudopodia, which we now note, and is consistent with our findings of mechanically balanced “in-groove” and “on-ridge” behavior and guided us to explore the influence of contractility in T cell motility.

2) Some of the main figures include rather idealized schematics that are not supported by the evidence within the figure, mainly for the reasons discussed above. For example, in Figure 1, panels B and D schematics show that the cells form lamellipodia and pseudopodia/blebs on the ridge in the grooves/3D collagen, respectively. These should be replaced by the primary data used to construct the figures in panel E (currently found in Supplementary Figure 1). It would be more appropriate for all the idealized schematics (Fig. 1B, C, and D; Fig. 2D, and Fig. 3C) to be condensed, moved to the end of the paper, and presented as a speculative models unless there are data included in the manuscript that directly supports them.

We have now added much more robust primary data to the manuscript that directly supports the phenotypes (e.g. Figures 2, Supplementary Figures 1, 2 and 3, and Movies 1 and 2). We note that the coexisting phenotypes and shifting balance of phenotypes can be difficult for some readers to visualize and as such the schematics serve to help the reader interpret this behavior at key points in the manuscript and often also present our hypotheses that are then supported by our primary findings. We, therefore, believe the schematics, in concert with our increased presentation of primary data, placed at the beginning of the Results section will help the readers to familiarize, visualize and effectively operate with the key concepts throughout the manuscript.

For instance, we have edited the Results text to better present our primary data and the additional data introduced during the revision such as:

“Examination of T cell behavior revealed that these platforms are sterically interactive as they concurrently allow for both T cell “in-groove” amoeboid invasiveness (i.e. sterically-interactive pseudopodia) into textures

and more mesenchymal-like “on-ridge” pseudopodia (i.e. lamellipodia) spreading behavior (see co-existing “on-ridge” and “in-groove” phenotypes in **Figure 1 and Supplementary Figures 1, 2, and 3**). Indeed, the interactions between activated primary human T cells and elastic nanotextures reveal a partial T cell interface invasiveness into the nanogrooves depth (i.e. “in-groove” plane; **Figure 1a, b and Supplementary Figures 1 and 2**). The remaining “noninvasive” part of the T cell-nanotexture interface is structured more like a flat lamellipodium spread atop of the nanotextured plane (i.e. more mesenchymal-like at the “on-ridge” level; **Figure 1b and Supplementary Figures 1, 2, and 3**). Note, we define the mesenchymal-like phenotype for T cells as lamellipodium-driven flat adhesive spreading atop the nanoridges, similar to that observed during migration on flat environments. We also note that inhibition of Arp2/3, a key regulator of lamellipodia dynamics, confirms the presence of lamellipodia in T cells as inhibition abolishes the lamellipodia, resulting in a spindle-like phenotype (**Supplementary Figure 2**)”

and

“...we hypothesized that disruption of MTs would enhance T cell migration across a range of mechanical environments. Indeed, MT disruption results in a shift towards more “in-groove” protrusions (i.e. tilts to the phenotype balance toward a more amoeboid phenotype; **Figure 2d and Supplementary Figures 1-3, and Movies 1 and 2**) and highly directed migration from contact guidance across all substrate stiffnesses on both ICAM1 and FN (**Figure 2a, b and Supplementary Figure 4a,b**)”.

3) Some additional evidence could also reinforce the conclusions leading to their overall model of RhoA activation triggering rapid amoeboid migration. Specifically, can the change in GEF-H1 localization be confirmed following nocodazole addition and can they demonstrate that the abundance of RhoA-GTP increases in the same conditions and that increase requires GEF-H1 expression?

We thank the reviewer for this insightful suggestion. To strengthen the conclusion that microtubule destabilization increases Rho activity and also that this activation is mediated by GEF-H1, we have performed additional experiments and also added references to key literature supporting this behavior. We now show in Supplementary Figures 6d and 6e that Rho activation is increased in T cells following treatment with Nocodazole and that this increase does not take place in GEF-H1 KO cells. We have edited the text to include the reference to key literature and present the new data:

“In this context, from our data on “2.5D” substrates and reports highlighting increased Rho-mediated contractility via GEF-H1 following microtubule destabilization^{38-40,43}, we hypothesized that MT destabilization” and “Consistent with our model, results indicate that MT destabilization significantly increases Rho activity ~2.5-fold and the mechanical rigidity as a function of robust cortical actomyosin contractility (**Figure 3 and Supplementary Figure 6**)” and “Consistent with our hypothesis, we observed an ~2-fold increase in active Rho following Nocodazole treatment that is abolished in GEF-H1 knockout cells (**Supplementary Figure 6**) and that under MT instability, migration is significantly decreased in GEF-H1 KO cells compared to control cells expressing Cas9”

4) Including a representative video of cell migration or protrusion dynamics for some of their key results seems like a great opportunity to directly show the reader the dramatic changes they are measuring.

We have added two additional movies (as well as time course images in **Supplementary Figure 3**) showing the different protrusion dynamics under the key conditions of Control vs. Nocodazole treatment. Along with new data in **Figure 2 and Supplementary Figure 1** and the New **Supplementary Figure 2 and 3**, we now robustly show the distinct balance shifts between phenotypes under distinct key conditions.

5) Lamellipodia are a type of pseudopodia, thus labelling the in-groove protrusions as pseudopodia is not particularly informative. If their further experiments can clearly show they are a bleb-based protrusion, the more specific label should be applied where appropriate.

We thank the Reviewer for this observation. We have edited the manuscript text to highlight that both types of protrusions are pseudopodia and have further edited the text to better define each behavior and highlight the balance of phenotypes and their changes under different conditions. For instance, the second paragraph of the Results section now reads “Examination of T cell behavior revealed that these platforms are sterically interactive as they concurrently allow for both T cell “in-groove” amoeboid invasiveness (i.e. sterically-interactive pseudopodia) into textures and more mesenchymal-like “on-ridge” pseudopodia (i.e. lamellipodia) spreading behavior (see co-existing “on-ridge” and “in-groove” phenotypes in **Figure 1 and Supplementary Figures 1, 2, and 3**). Indeed, the interactions between activated primary human T cells and elastic nanotextures reveal a partial T cell interface invasiveness into the nanogrooves depth (i.e. “in-groove” plane; **Figure 1a, b and Supplementary Figures 1 and 2**). The remaining “noninvasive” part of the T cell-nanotexture interface is structured more like a flat lamellipodium spread atop of the nanotextured plane (i.e. more mesenchymal-like at the “on-ridge” level; **Figure 1b and Supplementary Figures 1, 2, and 3**). Note, we define the mesenchymal-like phenotype for T cells as lamellipodium-driven flat adhesive spreading atop the nanoridges, similar to that observed during migration on flat environments. We also note that inhibition of Arp2/3, a key regulator of lamellipodia dynamics, confirms the presence of lamellipodia in T cells as inhibition abolishes the lamellipodia, resulting in a spindle-like phenotype (**Supplementary Figure 2**). This is consistent with carcinoma cell behavior where competitive dynamics between nanogrooves-guided directed invasive protrusions and less oriented “on-ridge” spreading can influence cell orientation and migration¹¹, and consistent with electron microscopy findings of T cell lamellipodia spreading on top of nanostructured surfaces^{29,30}”

6) How do the protrusions identified in the 2.5D model relate to each other in a 3D context, i.e. what is the physiological relevance of the “2.5 D” substrates? Can the T cells migrating in 3D collagen use both types of protrusions simultaneously?

We thank the reviewer for highlighting our need to provide better rationale for our platforms and our goals for using each system. We employed the “2.5D” model in order to parse out shifts in cell phenotypes and dynamics that are most often elusive in more complex 3D environments and often indiscernible on homogenous 2D surfaces. We have edited the opening paragraph of the Results section to better highlight this fact and the relevance of the “2.5D” systems. It now reads “We previously demonstrated that aligned stromal ECM in tumors, which has also been reported to guide tumor-infiltrating T cells^{1,2}, contains anisotropic nanostructured collagen with spacing and textures that can be effectively mimicked through nanotexturing technology^{8,11}. This allows us to impart relevant directional migration cues that are a useful compromise between two-dimensional (2D) and three-dimensional (3D) environments and allow us to characterize phenotypes that cannot be easily discerned in 3D environments and perform quantitative analysis on large cell numbers under defined conditions. Therefore, in order to first decipher how tumor architectures and mechanics influence T cell motility, we designed “2.5D” nanotextured platforms with defined mechanical rigidities and oriented architectures.”

Furthermore, on the “2.5D” substrates we observe co-existing phenotypes that we believe also exist in more complicated fully 3D environments. However, our goal here was to utilize the “2.5D” systems for high throughput and to develop strategies to perturb T cells and then test the impact of altering these structures or signals on T cell migration in more complex environments. While isolating key and simultaneous phenotypes in 3D in relation to engineered or endogenous ECM architecture is very exciting, and certainly something we have interest in, these studies are extremely complex. From our early investments in this area it is clear that

parsing out this very dynamic behavior in 3D is extremely complex and pushes the limits of our imaging platforms (e.g. super-resolution approaches capture key details but suffer during rapid events where full 3D behavior is needed and have limited imaging depth into 3D matrices or tumors while multiphoton and second harmonic generation approaches needed for imaging relatively deep into 3D matrices or tumors to also capture ECM information have more limited resolution, particularly during rapid scanning). As such, the “2.5D” platforms that mimic ECM texture have great value as they allow us to capture behaviors that are not initiated by flat surfaces and elusive in 3D. We have edited the text to better highlight that we are testing our findings from “2.5D” substrates in 3D, e.g. “As our nanotextured platforms were designed to capture key alignments and textures observed in tumor collagen fiber networks^{6,8,11}, and our data suggests parallels between invasive “in-groove” pseudopodia and T cell pseudopodial protrusions observed during migration through 3D collagen matrices¹², we next sought to test the hypothesis that disrupting microtubules can enhance 3D motility.”

7) To apply Laplace's law for spheres to calculate cellular hydrostatic pressure from tension the cell needs to be spherical. Given all of the T cell images in the manuscript show the cells are not spherical, how can the authors justify this approach to calculate hydrostatic pressure?

We thank the reviewer for highlighting our need to more clearly present the AFM assay. Indeed the T cells weakly adhered to glass dishes pre-coated with poly-L-lysine are spherical during the AFM testing in order to isolate behavior and perform necessary analysis. We have added images of the cells in this spherical state into Figure 3 (see **Figure 3**) and edited the figure text to better present this point. We have also edited the associated **Supplementary Figure** (now **Supplementary Figure 6**) to better present the assay and analysis.

8) What evidence is there the nocodazole and taxol treatments are depolymerizing and stabilizing microtubules in these experiments, respectively? Including movies that show what treatments do to protrusions and IF images that show how treatments rearrange microtubules could help to illustrate the effectiveness of the treatments.

We have endeavored to better highlight differences and changes in behavior following treatment with microtubule targeting agents. We have added STED imaging of microtubules in control conditions and after either Nocodazole or Taxol treatment (**Supplementary Figure 1d**). Also, in **Figure 2** and **Supplementary Figure 2** we show that Nocodazole results in loss of microtubule filaments (diffuse monomeric tubulin signal remains, also shown in **Supplementary Figure 1d**, which is consistent with previous findings by us and others), while Taxol treatment results in robust microtubule filament structures. Both behaviors are consistent with the extensive literature for these drugs. We have also added additional data with movies of control vs. Nocodazole treatment showing distinct phenotype behaviors and added a new **Supplementary Figure 2** showing differences in microtubules and protrusions associated with different T cell phenotypes for T cells on 2D and “2.5D” substrates treated with vehicle or Nocodazole. **Figure 2** and **Supplementary Figure 1** also shows differences in protrusions and phenotypes in the presence of microtubule targeting agents. Lastly, at the first mention of MT destabilization and stabilization we now highlight data showing differences in MTs, i.e. “To test a role for MTs in regulating phenotype in human T cells, we pharmacologically destabilized MTs with Nocodazole or stabilized MTs with Taxol and evaluated resulting “in-groove” invasiveness (MT destabilization and stabilization are shown in **Figure 2a** and **Supplementary Figures 1 and 2**). Perhaps surprisingly, in contrast to carcinoma cells, MT destabilization...”

9) *The emphasis on engineered cells in the title seems incongruent with the fact they are only used in the final figure. I suggest the authors think about using a title that more reflects the major focus/findings of the paper.*

We appreciate the reviewer's insight. We do feel that this is the direction of the manuscript and the rationale for our study to identify and test targets to alter T cell migration using engineered systems and an engineering design approach that ultimately culminates in the final figures.

RESPONSE TO REVIEWER 3

We thank the reviewer for their critical evaluation of our work and for their comment "*I commend the authors on their well thought-out, hypothesis driven experiments*". Through these constructive criticisms we were able to identify additional ways to better present our work and our key findings, and develop and analyze additional experiments to improve the work. A number of important points and questions were raised that we have endeavored to address:

General Comment 1: This is a very novel study of how the physical properties of T-cells and T-cell migration are controlled by microtubules. The role of microtubules in T-cell migration is not surprising. However, the elegant studies presented in this manuscript establish new approaches for researching the precise mechanisms involved. In spite of the strong and novel physicochemical analysis, the report is rather shallow in T-cell and cancer biology. While focusing on the intratumoral migration of T-cells, the manuscript does not address the barriers to T-cell infiltration into tumors nor the adhesion molecules and chemokines involved. Among the barriers to T-cell infiltration is the role of tumor vascularity. For example, chordoma and chondrosarcoma are two sarcomas that are both quite avascular. Because of this avascularity, patients who have either of these two sarcomas currently have no chemotherapy options. This study also does not consider the immunotolerant microenvironment of the tumor. The language used to describe the KPC mouse model is confusing and probably incorrect; for example, reference to the KPC mouse being a model of metastatic pancreatic cancer.

We thank the reviewer for their encouraging comments. We have edited the manuscript to better clarify the focus of our study. While our focus here is to address physical barriers to effective T cell infiltration, we have edited the text to better highlight the additional barriers (through text edits and citations) present in pancreatic ductal adenocarcinomas. For instance, the Introduction now reads "We test our findings in engineered 2D to 3D systems and in live tumor pancreatic ductal adenocarcinomas (PDA), a cancer that has a robust fibrotic and immunosuppressive stroma²³⁻²⁶, and is frequently characterized by very limited and/or heterogeneous distributions of cytotoxic T cells^{4,26-28}.", while the Discussion section includes the following text: "Stromal cell populations and the ECM surrounding cancerous cells in the pancreas are believed to be critically involved in tumor growth, metastasis, and resistance to therapy^{25,54}, and also limit effective antitumor immunity from T cells due to immunosuppression and fibrotic barriers^{2,4,5,23,26-28}. Yet, following specific strategies to alter the stroma, PDA becomes susceptible to immune checkpoint blockade^{27,28} and engineered T cells can overcome some of the physical and chemical barriers⁵⁵. Thus, therapies that rely on T cell function can be effective against PDA, and other solid tumors, in the right setting. Yet, one key obstacle to maximally effective T cells therapies in solid tumors is their inability to efficiently navigate through heterogeneous tumor microenvironments to sample the entire tumor volume within a functional time domain". In addition, we agree with the reviewer that vascularity and vascular architecture and function can likely influence T cell infiltration. Indeed, PDA is also hypovascular (see Olive et al., Science 2009; Provenzano et al., Cancer Cell, 2012), yet anti-tumor T cells responses can be effective in the right context (highlighted above) suggesting that, while this is a barrier, it is not the most prominent barrier. Lastly, regarding the genetically engineered KPC model, it is a well-established model of autochthonous PDA that quite faithfully follows the clinical spectrum, histologic progression, and metastatic distribution of the human disease (e.g. Hingorani et al., Cancer Cell 2005,

Provenzano et al., Cancer Cell 2012) and in our cases the mice harbored metastatic disease (>85% of KPC harbor metastatic disease in the later stages, e.g. Hingorani et al., Cancer Cell 2005, Provenzano et al., Cancer Cell 2012).

General Comment 2) Finally, the study provides no in-vivo evidence that microtubule destabilization is protective in cancer, even in a simple animal model. The authors also do not address the fact that tumor cells mimic lymphocyte migration characteristics to metastasize; hence their approach may enhance tumor metastasis.

Our goal here was to define approaches to enhancing T cell migration through complex environments. While microtubule stabilizing agents are one standard of care for PDA (i.e. Gemcitabine + Abraxane), the preclinical studies and limited trials using microtubule destabilizing agents also suggest benefit (Smith et al., Anticancer Drugs, 1995; Vergeylen and Kloppel, Virchows Archiv, 1995; Young et al., Clinical Cancer Research 2006; Capuchino et al., Tumori 2003). In addition, our previous work shows that microtubule destabilization decreases carcinoma cell motility (Tabdanov et al., Cell Reports 2018 and Nature Communications 2018), suggesting a simultaneous benefit of decreasing carcinoma motility while increasing T cell motility in tumors. While this could be an interesting approach (e.g. accurately timed Vinblastine treatment) we envision using the information presented in our study to specifically engineer T cells while not exposing carcinoma cells to non-standard of care perturbing factors. See also the response to Specific Comment 8.

General Comment 3) The manuscript is well written, the overall experimental design is sound, the data are clearly presented, and the interpretation of the results is rational. The discussion could be made easier to follow for readers who are not in the field.

We thank the reviewer for highlighting the need to increase accessibility of our work. We have edited the Discussion in order to clarify our points and more clearly present our conclusions. See also responses to Specific Comments 2 and 4.

Specific comments:

1) The introduction nicely summarizes the current understanding of T-cell migration, as well as areas where further investigation is needed, thus offering rationale for the subsequent work. While the final paragraph of the manuscript offers additional rationale, I would suggest adding a statement or two in this paragraph that highlights the overall hypothesis so that readers are reminded of the motivation for the experiments.

Encouraged by the reviewers suggestion we have added text to the Introduction to highlight our overall hypothesis: "Here, we hypothesized that by elucidating mechanisms governing T cell motility behavior in response to defined mechanical and architectural microenvironmental cues, we can further utilize these principles to rationally perturb T cells in order to enhance their migration through physically complex tumor microenvironments."

2) The results are well organized, and there is significant discussion within the results section, including introductory information as well as conclusions that the authors have drawn based on their results. The actual data itself becomes lost. I suggest revisions to this section to make their important findings more clear, in addition to simply referencing figures. For example, when discussing T-cell migration directionality on various nanotexture rigidities, the authors state that quantification per every 10 second step demonstrates significant differences in T-cell migration directionality. Providing these data and stats in the text would help distinguish

actual results from this experiment from the discussion offered in this section. A similar example could be given for each sub-section within the results section.

We have edited the Results to limit rationale, literature citation, and discussion to essential elements and increasingly highlighted the primary data. Furthermore, we have edited the text surrounding the reviewer's suggestion to read "Indeed, quantification of T cell migration directionalities per every 10 second-step across distinct nanotexture rigidities (16 to 50 kPa to >1,000 kPa) demonstrates significant differences in T cell migration directionality, with collectively stronger guidance along the 0° angle of deviation from the nanogrooves orientation on soft ($G'=16$ kPa) nanotextures, but more random and less persistent directionalities on stiffer nanotextures (**Figure 2f**). That is, more than 75% of the T cell population migrate within only $\pm 20^\circ$ of deviation from the nanogrooves axis on the soft ($G'=16$ kPa) nanotextures at any given moment of time. On the contrary, all stiffer nanotextures ($G'=50$ kPa or greater) induce only $\sim 50\%$ of T cell population migrating within $\pm 40^\circ$ of deviation range from the axis of the nanogrooves. Thus, the softer nanotextures present T cells to more robustly sense and conform their migration dynamics to the topography of their microenvironments." and also edited throughout the Results to present more primary data in the text.

3) The experimental design includes only a single dose of each of the agents Nocodazole and Taxol. Additional dose levels would strengthen the data set for this experimental section.

We appreciate the reviewer's interest in dose response behavior. Here, the dosing was chosen based on robust published works in the field utilizing these doses and our own previous studies. For our questions we sought to robustly destabilize or stabilize microtubules with relevant dosing. At our chosen doses the microtubules are either effectively destabilized or stabilized (Figures 2 and Supplementary Figure 2) and thus higher dosing would not produce a more robust effect. Much lower dosing may produce a less robust effect, however determining the thresholds, while interesting, are not likely to further address our primary questions. Further, within our experimental time windows single dose effects are well maintained, suggesting an experimentally effective dose and that additional doses are not required.

4) Since the results and discussion sections are separate sections, the interpretations and comparisons to published works in the results section should be moved to the discussion section.

In line with Comment 2, we have edited the Results to limit rationale, literature citation, and discussion to essential elements and increasingly highlighted the primary data.

5) Key experiments showing the migration of T-cells in the presence of pancreatic cancer cells should be repeated using primary cells, preferably derived from organoid cultures or Patient Derived Xenograft (PDX) models having different immunomodulatory profiles (e.g., expression of different chemo-attractant or immunosuppressive molecules).

We agree with the reviewer that these would be interesting experiments. We have obtained an IRB to obtain blood and tissue from pancreatic cancer patients scheduled to undergo surgical resection. However, prior to the pandemic, no patients had consented to the blood collection and we have not obtained large enough pieces of tumor to generate slice cultures that retain the physical features that are the focus of this study and sufficiently isolate T cells (which are extremely limited in $\sim 75\%$ of pancreatic adenocarcinomas). Since the pandemic, tissue is no longer being released and this will likely remain the case for this year and likely at least a portion of next year. Therefore we used the genetically engineered *KPC* model that is highly faithful to human PDA.

6) *It will be important to use cells isolated from pancreatic cancer patients for the initial characterization of the nano-platform and define the role of GEF-H1 in the migration of those cells.*

In line with the Comment 5, to date no patients have consented to blood draws and we have not obtained sufficient volumes of primary tissue / tumors to effectively isolate the necessary T cells.

7) *The rationale of looking at GEF-H1 is not clearly defined. Why do the authors focus on this guanine exchange factor (GEF)? There are other GEFs expressed in T-cells that can activate Rho. This should be clarified.*

We thank the reviewer for pointing out a need to better define our rationale for focusing on GEF-H1. We have edited the text to better present our rationale, including key references in the text highlighting the specific connection between microtubules, GEF-H1, and Rho activation, e.g. "...and reports highlighting increased Rho-mediated contractility via GEF-H1 following microtubule destabilization^{38 40-43}". We have also added data (Supplementary Figure 6) directly connecting microtubule destabilization with increased Rho activation and that Rho activity in this context is GEF-H1 dependent. To better present our rationale the text now reads:

"...we next sought to test the hypothesis that disrupting microtubules can enhance 3D motility. Three dimensional fibrous ECM networks with more random ECM organization (**Figure 3a**) pose a greater challenge to effective migration as cells move as a series of short contact guidance steps along fibers, or groups of fibers, with the need to re-orient in order to transition to another fiber configuration for guidance cues or through tortuous porous paths. In this context, from our data on "2.5D" substrates and reports highlighting increased Rho-mediated contractility via GEF-H1 following microtubule destabilization^{38 40-43}, we hypothesized that MT destabilization will promote 3D migration and is influencing increased Rho activity behavior via GEF-H1 (**Figure 3b**)..."

and

"Consistent with our model, results indicate that MT destabilization significantly increases Rho activity ~2.5-fold and the mechanical rigidity as a function of robust cortical actomyosin contractility (**Figure 3 and Supplementary Figure 6**)"

and

"Consistent with our hypothesis, we observed an ~2-fold increase in active Rho following Nocodazole treatment that is abolished in GEF-H1 knockout cells (**Supplementary Figure 6**) and that under MT instability, migration is significantly decreased in GEF-H1 KO cells compared to control cells expressing Cas9, with overall motility cut in half (**Figure 5a-c**)".

8) *The theoretical potential for enhancing metastatic tumor cell migration needs to be discussed either by published reference articles or additional experiments.*

In the context of systemic treatment with a microtubule destabilizing agent, which our data suggests would enhance T cell migration, potential effects on carcinoma cells would certainly need to be considered. We envision using the information presented in our study to specifically engineer T cells while not exposing carcinoma cells to the same perturbing factors. Yet, current data suggests that treatment with a microtubule destabilizing agent would in fact decrease carcinoma cell migration. We previously observed decreased directed motility in carcinoma cells with destabilized microtubules (Tabdanov et al., Cell Reports 2018 and Nature Communications 2018). Likewise, as summarized in an elegant review by Bouchet and Akhmanova (Journal of Cell Science 2017) as "*This result was later reproduced by nocodazole-induced microtubule disassembly in cells cultured in soft collagen gels compared to stiff 2D substrates (Rhee et al., 2007). In*

addition, treatment with nocodazole was shown to abolish fibroblast motility in 3D matrices (Doyle et al., 2009). Importantly, the dependence of pseudopod elongation on microtubules was also observed in cancer cells that display a mesenchymal morphology in 3D, such as MDA-MB-231 cells (Kikuchi and Takahashi, 2008; Oyanagi et al., 2012). High doses of paclitaxel or nocodazole blocked migration of these cells in soft collagen gels (Carey et al., 2015). In fact, even low doses of different microtubule-targeting agents (MTAs), which were insufficient to block cell division, impaired matrix invasion by MDA-MB-231 cells (Tran et al., 2009)". We have edited the following text in our report to highlight this point: "Perhaps surprisingly, in contrast to carcinoma cells, MT destabilization (+Nocodazole) profoundly increases T cell amoeboid "in-groove" invasiveness (area fraction>50%) across all nanotexture mechanical rigidities" and "...taxane agents requires additional scrutiny as these chemotherapies may in fact limit cytotoxic T cell migration into and through tumors. On the other hand, MT disruption can diminish carcinoma cell directed motility^{10,11} while promoting T cell migration and therefore manipulating the microtubule-associated signaling and/or contractility has the potential to significantly enhance T cell migration through ECM architectures found in tumors".

9) The report is written in a very difficult prose and contains much "jargon". It can surely be translated into more easy to read English. In its present format, it would be of interest to a rather limited audience and thus be more suitable for a specialized journal.

We have edited the manuscript to increase readability and further increase access for wide readership.

REVIEWER COMMENTS

Reviewer #1 (Remarks to the Author):

The revised manuscript by the Provenzano group has strengthened and clarified questions from the reviewers.

Reviewer #2 (Remarks to the Author):

While Tabdanov et al addressed several of my concerns and suggestions in their resubmission, the identity of the in-groove protrusions remains unresolved in my opinion. This ambiguity in their results could be resolved by adjusting their nomenclature to be more general to avoid excessive speculation and unwarranted conclusions. However, a major concept presented in their paper relies on the fact that they are able to measure the balance between amoeboid and mesenchymal migration modes in cells moving on grooved surfaces. It is not obvious to me that the manuscript would retain its novelty and/or impact without making this distinction between protrusion types. It is clear to me that concluding the in-groove protrusions represent an amoeboid, bleb-like protrusion is simply unsupported by the existing data. Without direct evidence that the T cells are migrating using distinct protrusions, a major conclusion of the manuscript relies solely on speculation. This remains a critical weakness that needs to be addressed either through additional experimentation as outlined below or avoiding excessive speculations within the results section. It is clear that the authors have discovered a clear link between microtubule stability, actomyosin contractility and cell speed in 2.5D and 3D environments even though their proposed mechanism is unconvincing.

Comments:

1. There are no data presented that clearly show the T cells are using distinct types of protrusions to migrate across the ridges versus in the grooves of the 2.5D model. This could include using This could be accomplished by showing the presence or absence of specific protein markers (such as Arp2/3 or cortactin in the case of lamellipodia), demonstrating the protrusion morphology and dynamics are distinct from one another, or showing the protrusion dynamics are can be distinguished by their sensitivity to select chemical inhibitors. The finding that both in-groove protrusions and on-ridge lamellipodia are disrupted by CK-666 treatment suggests that they are not distinct protrusions, for example. Without this experimental data, any conclusions that microtubule stability and contractility are regulating the types of protrusions formed and thereby cell speed rely on excessive speculation. It seems equally likely that the disruption of microtubules could be triggering an increase in traction force and adhesion that manifests as in-groove protrusions (see King Lam Hui and Arpita Upadhyaya, PNAS May 23, 2017 114 (21) E4175-E4183).
2. Several of the main figures include rather idealized schematics that are not supported by the evidence within the figure. These illustrations should be clearly labeled as presenting hypotheses or speculations. Otherwise it appears you are using these figures to illustrate your major findings, when in fact they are more speculative in nature. The author's credibility is reduced without this distinction.
3. The GEF-H1/RhoA pathway should be described in the introduction.

Reviewer #3 (Remarks to the Author):

I have reviewed the 3 general comments and the 8 specific comments that we made on the initial submission of this manuscript, and the authors' responses to those comments in this revised manuscript. The answers to our comments were appropriate and thorough. We are quite satisfied with them. In addition, the authors' answers to the comments by reviewers 1 and 2 address those

comments from our perspective. The entire revised manuscript is, from our perspective (review team 3), of high quality. We have no additional comments or suggestions on the revised manuscript. Thank you for giving us the opportunity to review this body of work.

RESPONSE TO REVIEWERS

Re: Article number NCOMMS-20-22556-T: “Engineering T cells to enhance 3D migration through structurally and mechanically complex tumor microenvironments”

We are pleased to submit a significantly revised version of our manuscript. The suggestions have helped to improve and clarify the work. We have endeavored to address all of the concerns raised. We have revised the text and added additional data and analysis. We believe these efforts have extended the rigor and impact of the work. We present below a detailed point-by-point response to the reviewers' questions and comments.

RESPONSE TO REVIEWERS 1 and 3

Reviewer 1 comment:

“The revised manuscript by the Provenzano group has strengthened and clarified questions from the reviewers.”

Reviewer 3 comment:

“I have reviewed the 3 general comments and the 8 specific comments that we made on the initial submission of this manuscript, and the authors' responses to those comments in this revised manuscript. The answers to our comments were appropriate and thorough. We are quite satisfied with them. In addition, the authors' answers to the comments by reviewers 1 and 2 address those comments from our perspective. The entire revised manuscript is, from our perspective (review team 3), of high quality. We have no additional comments or suggestions on the revised manuscript. Thank you for giving us the opportunity to review this body of work.”

Response: We thank the reviewers for their comments that the revisions have “*strengthened and clarified questions from the reviewers (R1)*” and that we have adequately addressed the comments from the previous review (R3) and “*The entire revised manuscript is, from our perspective (review team 3), of high quality. We have no additional comments or suggestions on the revised manuscript (R3)*”.

RESPONSE TO REVIEWER 2

General comment: *“While Tabdanov et al addressed several of my concerns and suggestions in their resubmission, the identity of the in-groove protrusions remains unresolved in my opinion. This ambiguity in their results could be resolved by adjusting their nomenclature to be more general to avoid excessive speculation and unwarranted conclusions. However, a major concept presented in their paper relies on the fact that they are able to measure the balance between amoeboid and mesenchymal migration modes in cells moving on grooved surfaces. It is not obvious to me that the manuscript would retain its novelty and/or impact without making this distinction between protrusion types. It is clear to me that concluding the in-groove protrusions represent an amoeboid, bleb-like protrusion is simply unsupported by the existing data. Without direct evidence that the T cells are migrating using distinct protrusions, a major conclusion of the manuscript relies solely on speculation. This remains a critical weakness that needs to be addressed either through additional experimentation as outlined below or avoiding excessive speculations within the results section. It is clear that the authors have discovered a clear link between microtubule stability, actomyosin contractility and cell speed in 2.5D and 3D environments even though their proposed mechanism is unconvincing.”*

We thank the reviewer for the opportunity to further expand and clarify our work. We have made a number of changes to the manuscript to better address the “in-groove” vs. “on-ridge protrusions”. Notably, we have edited the text to better highlight what are our hypotheses and conclusions and use a more general nomenclature (see Response to Specific Comment 1 below) and have also added new data (**Supplementary Figure 6**). We have also edited labels within the Figures and the Figure text. Along these lines, we find it is important to highlight that T cell amoeboid-mesenchymal dual dynamics are not binary and are not exclusive of each other, but coexist and we hypothesize that they compete during T cells migration dynamics on the biophysical level. We conclude that increased cortical contractility increases the ability to *initially* invade into the nanogrooves, but also note that steric interactions from “in-groove” protrusions does not preclude T cells from then developing adhesion in the nanogrooves (our experiments using *flat* 2D substrates versus nanotextures also helps us further parse out key behaviors). Likewise, in the first revision we extensively expanded our analysis of the “in-groove” vs. “on-ridge protrusions (edits and new data to **Figures 1 and 2**, new data in **Supplementary Figures 1-3**, and new **Movies 1-4**, where phase contrast images in **Supplementary Figure 3 and Movies 1-4** show the lamellipodia phenotype under control conditions and significantly reduced lamellipodium and more amoeboid-like behavior following microtubule destabilization in line with field standards using morphology evaluation). We have also now added additional data for inhibition of Formins which, consistent with our hypothesis, tilts the balance between the co-existing phenotypes (new **Supplementary Figure 6 – See also Response to Specific Comment 1 below**). A detailed response the Reviewer’s specific comments is included below:

1. *“There are no data presented that clearly show the T cells are using distinct types of protrusions to migrate across the ridges versus in the grooves of the 2.5D model. This could include using This could be accomplished by showing the presence or absence of specific protein markers (such as Arp2/3 or cortactin in the case of lamellipodia), demonstrating the protrusion morphology and dynamics are distinct from one another, or showing the protrusion dynamics are can be distinguished by their sensitivity to select chemical inhibitors. The finding that both in-groove protrusions and on-ridge lamellipodia are disrupted by CK-666 treatment suggests that they are not distinct protrusions, for example. Without this experimental data, any conclusions that microtubule stability and contractility are regulating the types of protrusions formed and thereby cell speed rely on excessive speculation. It seems equally likely that the disruption of microtubules could be triggering an increase in traction force and adhesion that manifests as in-groove protrusions (see King Lam Hui and Arpita Upadhyaya, PNAS May 23, 2017 114 (21) E4175-E4183).”*

We thank the Reviewer for highlighting our need to better clarify our rationale and findings. During the revisions we have added extensive additional data and analysis showing distinct protrusions at the “on-ridge” and “in-groove” levels. We have also here further edited the text (copied below) to better discuss the protrusions and our hypotheses and conclusion, and we have added new data using inhibition of Formins to provide more rigor to our hypothesis and conclusions. Notably we added movies of cells migrating with different phenotypes in the first revision and the phase contrast images in **Supplementary Figure 3** clearly show the lamellipodia phenotype under control conditions and significantly reduced lamellipodium during amoeboid behavior following microtubule destabilization in line with field standards using morphology evaluation. Unfortunately, to our knowledge, no clear distinct, binary, and definitive specific molecular marker exists to distinguish different types of protrusions. While the lamellipodia will be Arp2/3 etc. positive, so will other regions of the cells. As such, we have used field standards for morphology to define the difference and demonstrate that protrusion morphology and dynamics are distinct (noted in **Figures 1 and 2, Supplementary Figures 1-3**, and **Movies 1-4**). We do note that our hypothesis for how “in-groove” protrusions form is that the pseudopods are initiated via blebbing due to cortical contractility (we have edited text, copied below, to better highlight this) and note that our biophysical measures support this conclusion. We also note that imaging bleb dynamics on the nanoscale in the relevant time domains is beyond current imaging capabilities available in the

field. We have also now added data following inhibition of Formins which, consistent with our hypothesis, tilts the balance between the co-existing phenotypes by disabling the mesenchymal adhesion-based mechanotransduction between the contractile cytoskeleton and integrins, including LFA-1 (Tabdanov et al., *Integrative Biology*, 2015).. Lastly, we note that we observe increased “in-groove” protrusions under softer conditions, not stiffer, and following disruption of microtubules the T cell motility becomes indifference to substrate stiffness with highly directed migration at the same speed for all conditions, which is in contrast to increased traction force resulting in more “in-groove” protrusions. Below we expand on each these points and highlight additions and changes to the revised manuscript.

KEY CHANGES TO THE TEXT WHEN FIRST INTRODUCING THE COEXISTING PHENOTYPES (changes highlighted with blue font):

“Examination of T cell behavior revealed that these platforms are sterically interactive as they concurrently allow for both T cell “in-groove” invasiveness (i.e., sterically-interactive pseudopodia) into textures an “on-ridge” pseudopodia spreading behavior (see co-existing “on-ridge” and “in-groove” phenotypes in **Figure 1 and Supplementary Figures 1-3**). The “noninvasive” part of the T cell-nanotexture interface is structured more like a flat lamellipodium spread atop of the nanotextured plane (i.e., more lamellipodia-like mesenchymal behavior at the “on-ridge” level; **Figure 1a, Supplementary Figures 1-3, and depicted in Figure 1b**). Note, we define the mesenchymal-like phenotype for T cells as lamellipodium-driven flat adhesive spreading atop the nanoridges (**Supplementary Figure 3**), similar to that observed during migration on flat environments¹⁸ and consistent with electron microscopy findings of T cell lamellipodia spread on top of very stiff nanostructured surfaces^{29,30}. This is also consistent with carcinoma cell behavior where competitive dynamics between nanogrooves-guided directed invasive protrusions and less oriented “on-ridge” spreading can influence cell orientation and migration¹¹. Furthermore, we note that in activated primary T cells where we observe partial T cell invasiveness into the nanogrooves depth of elastic nanotextures (i.e., the “in-groove” plane), we do not observe any filopodia precursors or flat protrusion along the groove walls, but instead observe a rounded phenotype (**Figure 1a and Supplementary Figures 1 and 2**), suggesting a more amoeboid-like behavior. Thus, in our model...”

AND

“From a mechanobiology perspective, the fact that we did not observe any filopodia or flat protrusion structures along the walls in nanogrooves of any stiffness, that invadopodia are very small and distinct punctate circular structures³⁶, and that lamellipodia require larger dimensions than the nanogrooves here provide and do not undergo sub-micron bending³⁷⁻³⁹ suggests to us that “in-groove” amoeboid pseudopodia formation is primarily initiated by cytoplasm hydrostatic pressure⁴⁰ generated by high cortical actomyosin contractility. Cytoplasm pressure-driven off-cortex plasma membrane peeling can result in budding of the plasma membrane blebs that can be pseudopodal precursors⁴¹⁻⁴³, similar to depictions in **Figure 1b**, which then rebuild an actomyosin cortex in the new pseudopodia extensions⁴¹⁻⁴³. Therefore, we hypothesized that in-groove pseudopodia formation dynamics can be inherently antagonized by mechanical tension from the cell-substrate adhesion interface from the “on-ridge” pseudopodia, i.e., the mesenchymal-like T cell adhesion mechanisms (**Figure 1b,d**), similar to observations in carcinoma cells¹¹. This suggests that the mechanical properties of the substrate, and the related intracellular mechanics, would play a key role in governing directed motility of T cells. To explore this regulation, we first engineered nanotextures (“2.5D”) and nanolines (flat 2D lines) of different stiffnesses functionalized with ICAM1. Evaluation of activated human T cell dynamics on “2.5D” substrates demonstrates mechanoresponsive behavior with significantly decreased “in-groove” interactions (i.e., “in-groove” invasiveness) as nanotexture stiffness increases (**Figure 1e**), consistent with our hypothesis. In fact, increasing mechanical rigidity from $G'=16$ kPa through 50 kPa (polyacrylamide gel: PAAG) to $G'\gg 1$ GPa (polyurethane, i.e., pU) results in a significant and consistent drop of “in-groove” invasiveness (defined as

the invasive area percentage of cell-substrate interface projection) from ~60% to ~45% to ~20%, respectively (**Figure 1e**; **Supplementary Figure 1a-c**), demonstrating that the degree of T cell “in-groove” amoeboid-like invasiveness is regulated by variable mechanotransduction resulting from differences in the mechanical rigidity of the microenvironment. This observation suggests that as the substrate stiffness increases, the “on-ridge” lamellipodium-driven T cell-to-substrate adhesion interface strengthens via increased traction forces from the mechanosensitive adhesion signaling feedback (i.e., the mesenchymal-like portion of the cell consistent with reported T cell mechanoresponsive behavior^{18,44,45,46}). Thus, we conclude that the “on-ridge” mesenchymal adhesive cell-substrate interface competes with pseudopodal protrusions, i.e., invasive protrusion initiated by blebbing that is driven by cytoplasm hydrostatic pressure from high cortical contractility. Indeed, our data suggest that the mechanical conditions of the T cell environment can influence a redistribution of forces between high cortical contractility and on-ridge cell-substrate traction and therefore manipulating this balance can influence T cell phenotype and motility.

REGARDING THE COMMENTS:

“demonstrating the protrusion morphology and dynamics are distinct from one another” and “It seems equally likely that the disruption of microtubules could be triggering an increase in traction force and adhesion that manifests as in-groove protrusions (see King Lam Hui and Arpita Upadhyaya, PNAS May 23, 2017 114 (21) E4175-E4183).”

Distinct differences in “in-groove” and “on-ridge” protrusion morphology are shown in multiple ways in **Figures 1 and 2** and **Supplementary Figures 1 and 2**). Likewise, our bright-field comparative analysis of the T cells under control conditions vs Nocodazole clearly shows a suppression of the lamellipodium in Nocodazole (**Supplementary Figure 3** and **Movies 1-4**). Furthermore, our data does not support a rigidity-driven enhancement of traction-induced mesenchymal lamellipodium/filopodium invasive penetration into the grooves as an explanation for increased T cell invasiveness into the nanogrooves. Namely, we observe a decrease in cell invasiveness into the nanogrooves as the substrate stiffness increases (i.e., more “in-groove” invasion into softer substrates; **Figure 1e**). Instead, increasing substrate rigidity leads to the flattening of the T cell-nanotexture interface within the “on-ridge” plane. This is consistent with published reports displaying non-invasive flat lamellipodia in T cells on very stiff nanotextured surfaces (*via* electron microscopy; please see Chen, Z., Atchison, L., Ji, H. & Leong, K. W. Nanograting structure promotes lamellipodia-based cell collective migration and wound healing. *Conf. Proc. IEEE Eng. Med. Biol. Soc.* **2014**, 2916–2919 (2014).; Kwon, K. W. *et al.* Nanotopography-guided migration of T cells. *J. Immunol.* **189**, 2266–2273 (2012).). Here we have softer substrates where both “on-ridge” and “in-groove” behavior can simultaneously be present.

Along these lines, we find it is important to highlight that T cell amoeboid-mesenchymal dual dynamics are not binary and are not exclusive of each other, but coexist and compete for the resulting T cells migratory dynamics on the biophysical level. Importantly, we conclude that increased cortical contractility increases the ability to initially invade into the nanogrooves, but also note that steric interactions from “in-groove” protrusions does not preclude T cells from developing adhesion in the nanogrooves. Furthermore, our experiments using flat 2D substrates versus nanotextures helps us further parse out behaviors. For instance, T cells on flat 2D nanolines clearly show the suppression of the lamellipodium in Nocodazole and switching toward the amoeboid 3D shape-shifting T cell dynamics (**Supplementary Figures 2 and 5**). Note that loss of mesenchymal-like lamellipodial protrusions and switching towards a more amoeboid phenotype interactions leads to the loss of the optimal T cell-nanolines interactions on flat substrates, leading to a drop of the cell migration speed on flat substrates (**Supplementary Figure 5**). On the contrary, on “2.5D” substrates sterically interactive softer nanotextures favor “in-groove” dynamics, consequently displaying an increased T cell migration speed and migration tracks alignment along all nanotextures in Nocodazole (**Figure 2a-c**). Thus Nocodazole-induced growth does not support increased “in-groove” traction.

With specific regard to the elegant study by Hui *et al.* The investigator generated data utilizing the HIT3a antibody-coated surfaces that mimic the antigen-TCR cognate interactions via HIT3a antibody's binding to the CD3 ϵ subunit of the TCR complex. This type of adhesion is very informative but is principally distinct from integrin-based adhesion, as it represents the unique and non-mesenchymal type of cell adhesion, i.e., molecular recognition of the antigen. Importantly, in the primary T cells, TCR-antigen interactions do not induce lamellipodia while displaying a distinctively stronger adhesion than that of integrins (e.g., LFA-1-ICAM1 interactions). TCR-antigen adhesion features principally different cytoskeletal dynamics, displaying a dense, highly localized 3D mode of F-actin polymerization into the actively protruding and pushing force-generating (i.e., TCR foci) that is distinct from integrin-based LFA-1-ICAM1 adhesion (e.g., see J. Husson, K. Chemin, A. Bohineust, C. Hivroz, N. Henry, *PLoS One* **2011**, 6, e19680). To this effect, in order to recapitulate a proper, circular lamellipodium-based immune synapse it is a routine requirement to introduce ICAM1 along with TCR-activating ligands (please, see Tabdanov, E. et al. Micropatterning of TCR and LFA-1 ligands reveals complementary effects on cytoskeleton mechanics in T cells. *Integr. Biol.* 7, 1272–1284 (2015); S. Kumari, D. Depoil, R. Martinelli, E. Judokusumo, G. Carmona, F. B. Gertler, L. C. Kam, C. V. Carman, J. K. Burkhardt, D. J. Irvine, M. L. Dustin, *Elife* **2015**, 4.; K. Shen, V. K. Thomas, M. L. Dustin, L. C. Kam, *Proc. Natl. Acad. Sci. U. S. A.* **2008**, 105, 7791.). Here we are using ICAM1 and ECM substrates, limiting the ability to make direct comparisons to behavior from TCR based adhesion.

REGARDING THE ADDITION OF NEW DATA:

To further test our hypothesis of forces from the “on-ridge” lamellipodia-like adhesion competing with formation of “in-groove” protrusions we inhibited Formins as described in the modified text:

“Furthermore, we further tested our hypothesis using inhibition of Formins, where inhibition does not substantially inhibit lamellipodia spreading, but are needed for force transmission between integrin adhesions in the lamella and the substrate^{47–51}, and observed a profound shifts in the phenotype balance toward a more “in-groove” T cell phenotype (**Supplementary Figure 6**). On stiff substrates this results in significantly greater directed migration (**Supplementary Figure 6**), which is consistent with our hypothesis that forces from the mesenchymal-like “on-ridge” level compete with cortical contractile forces that drive the phenotype balance more toward the amoeboid phenotype. Thus, we conclude that “in-groove” invasiveness resulting from high cortical contractility and confined pseudopodia promotes more efficient directed migration. Therefore, we suggest a mechanistic model for efficient T cell directed migration where “in-groove” invasive pseudopodal protrusions sterically guide and drive T cell migration along the nanogrooves (**depicted in Figure 2e**)...”. Thus, this data clearly shows that tilting the balance away from “on-ridge” contractility shifts the balance toward the more amoeboid-like phenotype that enhances directed T cell migration.

2. Several of the main figures include rather idealized schematics that are not supported by the evidence within the figure. These illustrations should be clearly labeled as presenting hypotheses or speculations. Otherwise, it appears you are using these figures to illustrate your major findings, when in fact they are more speculative in nature. The author's credibility is reduced without this distinction.

We have edited the text and the Figures to highlight how each schematic is presenting a hypothesis or summarizing and/or depicting a conclusion. For instance, we have edited labels within the figures and in the edited text copied above we note what is hypothesized and depicted. Likewise, related to Figure 2 we have edited the text to read “Therefore, we suggest a mechanistic model for efficient T cell directed migration where “in-groove” invasive pseudopodal protrusions sterically guide and drive T cell migration along the nanogrooves (**depicted in Figure 2e**)”.

3. The GEF-H1/RhoA pathway should be described in the introduction.

We have added text to the Introduction to summarize the key role of GEF-H1 as related to the current study. The text now reads “This microtubule-tractility axis can be confirmed and captured by using CRISPR technology to genome engineer T cells lacking GEF-H1, a Rho guanine exchange factor that can increase Rho activity following microtubule destabilization^{38 40-43}. Knockout of GEF-H1 in the context of microtubule instability...”

REVIEWERS' COMMENTS

Reviewer #2 (Remarks to the Author):

The authors have satisfactorily addressed my remaining concerns with the manuscript. I think there is some very fascinating cell biology that is yet to be resolved regarding exactly what the in-groove protrusions represent. I recognize the author's hypothesis that they represent an amoeboid, bleb-like protrusion remains an intriguing possibility (a possibility that is strengthened by their new supplemental figure 6). I appreciate that they have modified their language throughout the text to keep open the possibility that the in-groove structures may not represent a bleb-based protrusion. I look forward to the publication of their very provocative and interesting findings.

RESPONSE TO REVIEWERS

Re: Article number NCOMMS-20-22556-B: "Engineering T cells to enhance 3D migration through structurally and mechanically complex tumor microenvironments"

We are pleased to submit a revised version of our manuscript. The suggestions have helped to improve and clarify the work. We have endeavored to address all of the concerns raised. We present below a detailed point-by-point response to the reviewers' questions and comments.

RESPONSE TO REVIEWERS 2

Reviewer #2 Comment:

The authors have satisfactorily addressed my remaining concerns with the manuscript. I think there is some very fascinating cell biology that is yet to be resolved regarding exactly what the in-groove protrusions represent. I recognize the author's hypothesis that they represent an amoeboid, bleb-like protrusion remains an intriguing possibility (a possibility that is strengthened by their new supplemental figure 6). I appreciate that they have modified their language throughout the text to keep open the possibility that the in-groove structures may not represent a bleb-based protrusion. I look forward to the publication of their very provocative and interesting findings.

Response: We thank the reviewers for their comments that have helped strengthen the manuscript.